# Kronecker Mask and Interpretive Prompts are Language-Action Video Learners

**Jingyi Yang**[1*], **Zitong Yu**[23*], **Xiuming Ni**[4], **Jia He**[4], **Hui Li**[1†]

[1]University of Science and Technology of China,  [2]Great Bay University,
[3]Dongguan Key Laboratory for Intelligence and Information Technology,
[4]Anhui Tsinglink Information Technology Co.,Ltd.
`yangjingyi@mail.ustc.edu.cn`, `yuzitong@gbu.edu.cn`,
`{nixm,hejia}@tsinglink.com`, `mythlee@.ustc.edu.cn`

## Abstract

Contrastive language-image pretraining (CLIP) has significantly advanced image-based vision learning. A pressing topic subsequently arises: how can we effectively adapt CLIP to the video domain? Recent studies have focused on adjusting either the textual or visual branch of CLIP for action recognition. However, we argue that adaptations of both branches are crucial. In this paper, we propose **CLAVER**: a **C**ontrastive **L**anguage-**A**ction **V**ideo Learn**er**, designed to shift CLIP's focus from the alignment of static visual objects and concrete nouns to the alignment of dynamic action behaviors and abstract verbs. Specifically, we introduce a novel Kronecker mask attention for temporal modeling. Our tailored Kronecker mask offers three benefits 1) it expands the temporal receptive field for each token, 2) it serves as an effective spatiotemporal heterogeneity inductive bias, mitigating the issue of spatiotemporal homogenization, and 3) it can be seamlessly plugged into transformer-based models. Regarding the textual branch, we leverage large language models to generate diverse, sentence-level and semantically rich interpretive prompts of actions, which shift the model's focus towards the verb comprehension. Extensive experiments on various benchmarks and learning scenarios demonstrate the superiority and generality of our approach. Code is available at https://github.com/yjyddq/CLAVER.

## 1 Introduction

Video action recognition has long been a representative topic in video understanding. Over the past decade, there has been a continuous pursuit of learning spatiotemporal representations, giving rise to diverse architectures, such as traditional two-stream networks Simonyan & Zisserman (2014); Wang et al. (2016); Zhou et al. (2018); Karpathy et al. (2014); Xie et al. (2024), 3D convolutional neural networks Carreira & Zisserman (2017); Feichtenhofer (2020); Feichtenhofer et al. (2019); Hara et al. (2017); Qiu et al. (2017); Tran et al. (2015; 2018); Wang et al. (2018); Xie et al. (2018), and Video Vision Transformers Arnab et al. (2021); Bertasius et al. (2021); Fan et al. (2021); Liu et al. (2022); Patrick et al. (2021); Zhao et al. (2022); Li et al. (2022a); Yan et al. (2022). Recently, there has been increasing interest in leveraging visual-language models (VLMs) like CLIP Radford et al. (2021), Florence Yuan et al. (2021), and ALIGN Jia et al. (2021) for various video tasks, owing to the superior generalization abilities of these models. Several studies Wang et al. (2021); Lin et al. (2022); Ni et al. (2022); Ju et al. (2022); Rasheed et al. (2023); Tu et al. (2023); Chen et al. (2023) have devoted to adapt the CLIP for video action recognition, but they often focus on adjusting a single branch. According to predecessor studies, transferring CLIP from the image domain to the video domain involves two key considerations: 1) how to perform effective temporal modeling. 2) how to design suitable text descriptions for verb understanding that align with rich text semantics in the VLM's pre-training dataset. We argue that addressing both issues simultaneously is crucial.

---

[*]Equal Contribution
[†]Corresponding Author

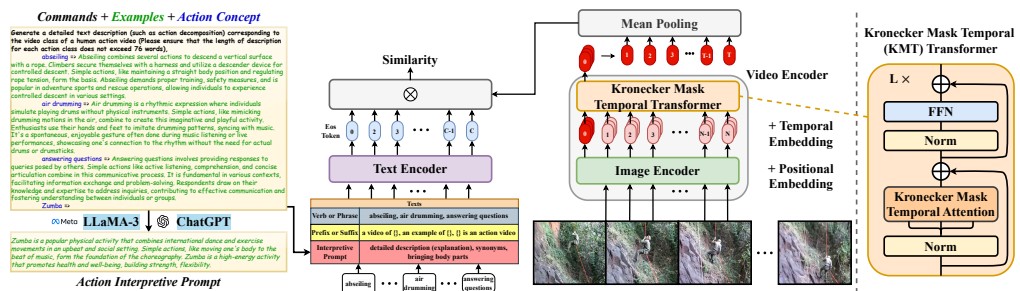

Figure 1: An overview of **CLAVER**. (**Right**) Image encoder and KMT transformer are assembled as a video encoder. (**Left**) How to get the interpretive prompts for actions.

In addressing the issue 1), several studies Wang et al. (2021); Ju et al. (2022); Chen et al. (2023); Rasheed et al. (2023) implement straightforward and simple strategies such as mean pooling or 1D-temporal convolution across the temporal dimension, or employing temporal attention among class tokens. X-CLIP Ni et al. (2022) and CLIP-ViP Xue et al. (2022) introduce extra tokens for cross-frame communication. Alternatively, some studies Lin et al. (2022); Tu et al. (2023) engineer tailored modules. In our work, we aim to elucidate the distinctions and intrinsic correlations between space and time, as well as design more general spatiotemporal modeling approaches.

Regarding the issue 2), some studies Hendricks & Nematzadeh (2021); Thrush et al. (2022) indicate that VLMs tend to focus on the correspondence between visual objects and nouns rather than action behaviors and verbs. We consider the essential gap is that visual objects are static and presented in lower dimensions, and nouns are concrete and easily understandable. However, action behaviors are dynamic that presented in higher dimensions, and verbs are intricate and abstract. Several existing methods Ni et al. (2022); Lin et al. (2022); Rasheed et al. (2023); Tu et al. (2023) use verbs or phrases as direct text descriptions. ActionCLIP Wang et al. (2021) integrates prompt templates to expand verbs or phrases into sentences. Ju et al. Ju et al. (2022) propose trainable continuous prompts to construct virtual prompt templates. However, these methods do not address aforementioned issues in essence. Alternatively VFC Momeni et al. (2023) and MAXI Lin et al. (2023) consider leveraging large language models (LLMs) to provide positive and negative text samples for contrastive learning or multiple instance learning, while ASU Chen et al. (2023) presents the concept of semantic units to supplement the semantic information of action labels.

To address these issues, we propose a **C**ontrastive **L**anguage-**A**ction **V**ideo Learn**er** (**CLAVER**, Fig. 1) to efficiently adapt the CLIP for video action recognition. Specifically, for the issue 1), we first obtain the frame-level visual representation from the image encoder, then apply tailored Kronecker mask for temporal modeling with a wider temporal receptive field to establish long-range and wide-range dependencies among frames, while mitigating spatiotemporal homogenization. Additionally, we reveal the intrinsic correlations between space and time from the perspective of Kronecker mask attention. Regarding the issue 2), we leverage LLMs to effectively generate diverse, sentence-level, and semantically rich interpretations of actions, augmenting text descriptions during training and testing. This approach allows the text descriptions to be presented in a more flexible, sentence-level form during inference. In summary, our main contributions are four-fold:

- We propose the **C**ontrastive **L**anguage-**A**ction **V**ideo Learn**er** (**CLAVER**) to adapt both the visual and textual branches, efficiently shifting the alignment in CLIP from visual objects and nouns to action behaviors and verbs.

- We propose the Kronecker mask temporal attention and Kronecker mask causal temporal attention for temporal modeling, aiming to capture the long-range and wide-range dependencies among frames with spatiotemporal heterogeneity.

- We introduce interpretive prompts of actions to facilitate the alignment of action behaviors and verbs, thereby improving zero-shot and few-shot generalization capabilities.

- Extensive qualitative and quantitative experiments demonstrate the effectiveness of CLAVER. Our method achieves superior or competitive performance on Kinetics-400 and Kinetics-600 under fully-supervised scenario, and on HMDB-51 and UCF-101 under zero-shot, few-shot scenarios.

## 2 RELATED WORK

**Video Recognition.** Among early video recognition methods, 3D convolution is widely employed Qiu et al. (2017); Tran et al. (2015; 2018); Xie et al. (2018); Feichtenhofer et al. (2019); Feichtenhofer (2020); Yu et al. (2021). Some studies Qiu et al. (2017); Tran et al. (2018); Xie et al. (2018) propose to factorize convolutional operations across spatial and temporal dimensions, while others design the specific temporal modules to embed them into 2D CNNs Li et al. (2020b); Lin et al. (2019); Liu et al. (2021). Over the past few years, there has been an influx of transformer-based video works Arnab et al. (2021); Neimark et al. (2021); Bertasius et al. (2021); Fan et al. (2021); Liu et al. (2022); Yan et al. (2022); Li et al. (2022a), demonstrating promising performance. For example, some methods Arnab et al. (2021); Neimark et al. (2021); Girdhar & Grauman (2021) adopt a factorized encoder structure for spatial-temporal fusion. Alternatively, another family employ a factorized attention structure. Such as TimeSformer Bertasius et al. (2021), ViViT Arnab et al. (2021), and ATA Zhao et al. (2022) which proposes alignment-guided tepmoral attention. While Video Swin Transformer Liu et al. (2022) employs 3D window attention. The representative attention calculation forms in these works can be summarized into joint attention and factorized (divided) spatiotemporal attention. In this paper, to make minimal modifications to the original structure, we adopt a factorized encoder structure for video stream.

**Visual-language Representation Learning.** Visual-textual multi-modality is a hot topic in recent years. Several studies based on masked image modeling (MIM) have achieved commendable performance Li et al. (2020a); Lu et al. (2019); Su et al. (2019); Tan & Bansal (2019). There are also efforts focused on video-language representation learning Miech et al. (2019); Sun et al. (2019a;b); Zhu & Yang (2020); Xu et al. (2021). Concurrently, contrastive language-image pretraining Radford et al. (2021); Jia et al. (2021); Yuan et al. (2021) achieved remarkable progress, particularly in demonstrating impressive zero-shot generalization capacities. CLIP Radford et al. (2021) is one of the most representative works, with numerous follow-up studies have explored to adapt it for downstream tasks. For example, object detection, semantic segmentation, video retrieval and captioning, etc.Gu et al. (2021); Vinker et al. (2022); Li et al. (2022b); Luo et al. (2022); Xu et al. (2022). Additionally, there are also many applications in the video action recognition Wang et al. (2021); Ni et al. (2022); Lin et al. (2022); Ju et al. (2022); Pan et al. (2022); Rasheed et al. (2023); Chen et al. (2023); Tu et al. (2023); Lin et al. (2023); Momeni et al. (2023); Yang et al. (2023). For instance, ViFiCLIP Rasheed et al. (2023) aim to minimize modifications to original models and facilitate efficient transfer, while Chen Ju et al. Ju et al. (2022) suggest optimizing a few prompt vectors for adapting CLIP to various video understanding tasks. X-CLIP Ni et al. (2022) proposes an efficient cross-frame attention module. ILA Tu et al. (2023) designs implicit mask-based alignment to align features of two adjacent frames and EVL Lin et al. (2022) proposes a image encoder and video decoder structure. Regrading to the language branch, most previous works directly use verbs or phrases that lack rich semantics which overlook the importance of semantics. With the advancement of large language models like the GPT-3 Brown et al. (2020), PaLM Chowdhery et al. (2023), LLaMAs Touvron et al. (2023a;b); Abhimanyu Dubey et al. (2024) and ChatGPT OpenAI (2024). LLMs can replace manual labor Chen & Huang (2021); Qian et al. (2022) and automatically generate texts that meet human expectations to benefit visual-textual learning. For example, LaCLIP Fan et al. (2024) employs LLMs to rewrite text descriptions associated with each image for text augmentation. VFC Momeni et al. (2023) and MAXI Lin et al. (2023) leverage LLMs to generate positive and negative texts with diversity for language-video learning.

## 3 METHODOLOGY

In Sec. 3.1, we overview our proposed contrastive language-action video learner architecture. Then, we elaborate on the detail of the Kronecker mask attention in Sec. 3.2. Finally, we present the technique details of action interpretive prompt in Sec. 3.3.

### 3.1 OVERVIEW

Our contrastive language-action video learner architecture is illustrated in Fig. 1. We utilize a video encoder to obtain video representations, comprising two transformers-based components: an image encoder (ViT) from CLIP, a Kronecker mask temporal transformer. The text encoder

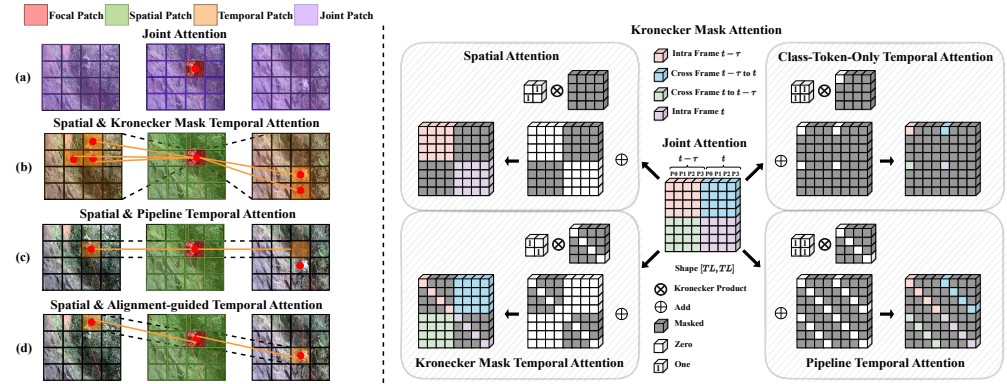

Figure 2: (**Left**) Red indicates the currently focal patch, green patches are visible in spatial attention, orange patches are visible in temporal attention, purple patches are visible in joint attention. (**Right**) Kronecker mask attention: Several attentions can be seen as employing tailored Kronecker masks for joint attention.

aims to align text representations with the video representations. Concretely, given a video clip $V = [v_0, \cdots v_t, \cdots v_{T-1}] \in \mathbb{R}^{T \times H \times W \times 3}, v_t \in \mathbb{R}^{H \times W \times 3}$ and corresponding text descriptions $C = [c_0, \cdots c_m, \cdots c_{M-1}] \in \mathbb{R}^{M \times N}, c_m \in \mathbb{R}^N$, where $T, H, W$ are the number of frames, height, width, respectively, $M$ is the number of diverse text descriptions (share the same central concept) for an action category, $N$ is the max sequence length. We feed texts $C$ into the text encoder $f_{\theta_C}(\cdot)$ to obtain text representations $\mathbf{C} = [\mathbf{c}_0, \cdots \mathbf{c}_{M-1}]$. For the video stream, firstly, we input the video clip $V$ to the image encoder $f_{\theta_I}(\cdot)$ to obtain frame-level representations $\mathbf{I}_t$.

$$\mathbf{I}_t = f_{\theta_I}(\text{PE}(v_t) + \mathbf{e}^{pos}), \quad \mathbf{C} = f_{\theta_C}(C), \tag{1}$$

where $\text{PE}(\cdot)$ is the patch embedding. Each frame is split into $L = \frac{H}{P} \times \frac{W}{P}$ patches, $L$ is the number of patches, $P$ is the patch size, $\mathbf{e}^{pos}$ is the absolute positional embedding, $\text{PE}(v_t) + \mathbf{e}^{pos} = [v_{t,0} + \mathbf{e}_0^{pos}, \text{PE}(v_{t,1}) + \mathbf{e}_1^{pos}, \cdots, \text{PE}(v_{t,l}) + \mathbf{e}_l^{pos}, \cdots, \text{PE}(v_{t,L}) + \mathbf{e}_L^{pos}] = [\mathbf{z}_{t,0}, \cdots, \mathbf{z}_{t,l}, \cdots, \mathbf{z}_{t,L}]$, $v_{t,0}$ is the class token. $\mathbf{I}_t = f_{\theta_I}([\mathbf{z}_{t,l}]_{t \in T, l \in L+1}) = [\mathbf{I}_{t,0}, \cdots, \mathbf{I}_{t,l}, \cdots, \mathbf{I}_{t,L}]$.

Then, we add absolute temporal embedding $\mathbf{e}^{tem}$ to the $\mathbf{I}_t, t \in T$ and feed them into the Kronecker mask temporal transformer $f_{\theta_V}(\cdot)$. Finally, by selecting the class token from each frame and averaging them, we obtain a video representation $\mathbf{v}$ with the same dimension as $\mathbf{c}_m, m \in M$.

$$\mathbf{V} = f_{\theta_V}([\mathbf{I}_t]_{t \in T} + \mathbf{e}^{tem}), \quad \mathbf{v} = \text{Avg}([\mathbf{V}_{t,0}]_{t \in T}), \tag{2}$$

where $\mathbf{V} = f_{\theta_V}([\mathbf{I}_t]_{t \in T} + \mathbf{e}^{tem}) = f_{\theta_V}([\mathbf{I}_t + \mathbf{e}_t^{tem}]_{t \in T}) = f_{\theta_V}([[\mathbf{I}_{0,0} + \mathbf{e}_0^{tem}, \cdots, \mathbf{I}_{0,L} + \mathbf{e}_0^{tem}], \cdots, [\mathbf{I}_{t,0} + \mathbf{e}_t^{tem}, \cdots, \mathbf{I}_{t,L} + \mathbf{e}_t^{tem}], \cdots, [\mathbf{I}_{T-1,0} + \mathbf{e}_{T-1}^{tem}, \cdots, \mathbf{I}_{T-1,L} + \mathbf{e}_{T-1}^{tem}]]) = [[\mathbf{V}_{0,0}, \cdots, \mathbf{V}_{0,L}], \cdots, [\mathbf{V}_{t,0}, \cdots, \mathbf{V}_{t,L}], \cdots, [\mathbf{V}_{T-1,0}, \cdots, \mathbf{V}_{T-1,L}]]$, $\text{Avg}(\cdot)$ is the average pooling function. Our optimization goal is to maximize the cosine similarity between video $\mathbf{v}$ and its corresponding texts $\mathbf{c}_m \in \mathbf{C}$ representations:

$$\text{sim}(\mathbf{v}, \mathbf{c}_m) = \frac{\langle \mathbf{v}, \mathbf{c}_m \rangle}{\|\mathbf{v}\| \cdot \|\mathbf{c}_m\|}. \tag{3}$$

### 3.2 KRONECKER MASK ATTENTION

For an image $\in \mathbb{R}^{H \times W \times 3}$, it is first split into patches and then flattened into a token sequence after patch embedding. The resulting feature shape is $(L, D)$, $L = \frac{H}{P} \times \frac{W}{P}$, where $D$ is the hidden dimension. This process is a standard operation in ViT Dosovitskiy et al. (2020), denoted as Spatial Attention (SA):

$$\mathbf{Z}_{(L,D)} = \text{Softmax}(\mathbf{Q}_{(L,D)}\mathbf{K}_{(L,D)}^{\mathbf{T}}/\sqrt{D})\mathbf{V}_{(L,D)}, \tag{4}$$

where $\mathbf{Q}$, $\mathbf{K}$, $\mathbf{V}$ represent the query, key, value matrices, respectively.

For a video $\in \mathbb{R}^{T \times H \times W \times 3}$, $T$ is the number of frames, previous studies Arnab et al. (2021); Bertasius et al. (2021); Neimark et al. (2021); Guo et al. (2021); Tong et al. (2022); Feichtenhofer et al. (2022) typically employ either joint attention Bertasius et al. (2021); Feichtenhofer et al. (2022) or

factorized (divided) attention Bertasius et al. (2021); Arnab et al. (2021). Joint attention flattens a video into a longer token sequence, resulting in a feature shape of $(T \times L, D)$. This token interaction mode is illustrated in Fig. 2 **Left** (a). Joint attention encounters spatiotemporal homogenization issues, where random shuffling of tokens not affecting (or slightly) the final pooling result.

In contrast, factorized attention factorize joint attention into spatial attention and temporal attention to avoid spatiotemporal homogenization. They utilize the same spatial attention as Eqn. 4, while the temporal attention varies. Two common temporal attentions are pipeline temporal attention Arnab et al. (2021); Bertasius et al. (2021) and class-token-only temporal attention Arnab et al. (2021); Ni et al. (2022); Wang et al. (2021). Their feature shapes are $(L, T, D)$ and $(T, D)$, respectively. Pipeline temporal attention has a limited temporal receptive field, which is limited to a fixed time pipeline (tube), hence the term "pipeline temporal attention", as shown in Fig. 2 **Left** (c). It has limited scope of capture dynamic information since objects of interest do not always appear in the same 2D location across frames. Although ATA Zhao et al. (2022) utilizes alignment techniques to bend the time pipeline to capture dynamic objects, it still has a limited temporal receptive field, in Fig. 2 **Left** (d). Similarly, class-token-only temporal attention retains only the class token time pipeline and discard others, which both face limited receptive field and may discard lots of potentially valuable information. It is worth noting that in pipeline temporal attention, mean pooling is necessary across all tokens, otherwise, it is equivalent to class-token-only temporal attention.

To address aforementioned drawbacks, we propose Kronecker Mask Temporal Attention (KMTA). Specifically, we allow each patch (token) at timestamp $t$ can interact with all other patches (tokens), excluding those sharing the same timestamp $t$, as illustrated in Fig. 2 **Left** (b). Compared to pipeline temporal attention, KMTA expands the temporal receptive field width of each token. KMTA can be achieved through joint attention incorporated a Kronecker mask, as shown in Fig. 2 **Right** (left down). Additionally, KMTA alleviates the impact of spatiotemporal homogenization due to the presence of the Kronecker mask. The trick for obtaining the Kronecker mask is Kronecker product $\otimes$:

$$\mathbf{A}_{m \times n} \otimes \mathbf{B}_{p \times q} = \begin{bmatrix} a_{11}\mathbf{B} & a_{12}\mathbf{B} & \cdots & a_{1n}\mathbf{B} \\ a_{21}\mathbf{B} & a_{22}\mathbf{B} & \cdots & a_{2n}\mathbf{B} \\ \vdots & \vdots & \ddots & \vdots \\ a_{m1}\mathbf{B} & a_{m2}\mathbf{B} & \cdots & a_{mn}\mathbf{B} \end{bmatrix}_{mp \times nq} \tag{5}$$

Eqn. 5 is the definition of $\otimes$, where $\mathbf{A} \in \mathbb{R}^{m \times n}$, $\mathbf{B} \in \mathbb{R}^{p \times q}$. Thus, it is referred to as the Kronecker mask. Kronecker Mask Temporal Attention (KMTA) can be formulated as:

$$\mathbf{M}_{(T \times L, T \times L)} = [\mathbf{I}_{(T,T)} \otimes (\mathbf{J}_{(L,L)} - \mathbf{I}_{(L,L)})]_{1==-\inf}, \tag{6}$$

$$\mathbf{Z}_{(T \times L, D)} = \text{Softmax}(\mathbf{Q}_{(T \times L, D)} \mathbf{K}^{\mathbf{T}}_{(T \times L, D)} / \sqrt{D} + \mathbf{M}_{(T \times L, T \times L)}) \mathbf{V}_{(T \times L, D)}, \tag{7}$$

where $\mathbf{I}_{(-,-)}$ is an identity matrix, $\mathbf{J}_{(-,-)}$ is an all-ones matrix, $[\quad]_{1==-\inf}$ means replacing 1 in the matrix with negative infinity ($-\inf$), and $\mathbf{M}_{(T \times L, T \times L)}$ is the Kronecker mask. A transformer equipped with KMTA is referred to as a Kronecker Mask Temporal (KMT) transformer.

In fact that both the spatial and temporal attention we mentioned above can be derived by combining a tailored Kronecker mask with joint attention. Therefore, we collectively refer to them as Kronecker Mask Attention, as depicted in Fig. 2 (**Right**). The Kronecker mask serves as a prior spatiotemporal heterogeneity inductive bias. Spatial attention allows intra-frame interactions but blocks inter-frame interactions, while Kronecker mask temporal attention allows inter-frame interactions but blocks intra-frame interactions, exhibiting a spatiotemporal structural complementarity.

Moreover, we design another type of Kronecker mask for temporal modeling, known as Kronecker mask causal temporal attention (KMCTA) that aims to alleviate the low-rank bottleneck, as shown in Fig. 3. The mask $\tilde{\mathbf{M}}_{(T \times L, T \times L)}$ of KMCTA can be formulated as:

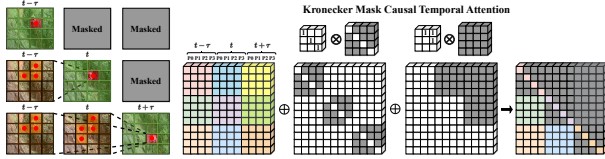

Figure 3: Kronecker mask causal temporal attention.

$$\tilde{\mathbf{M}}_{(T \times L, T \times L)} = [\mathbf{I}_{(T,T)} \otimes (\mathbf{J}_{(L,L)} - \mathbf{I}_{(L,L)}) + (\mathbf{U}_{(T,T)} - \mathbf{I}_{(T,T)}) \otimes \mathbf{J}_{(L,L)}]_{1==-\inf}, \tag{8}$$

where $\mathbf{U}_{(-,-)}$ is an upper triangular matrix, and all elements of the upper triangle are 1. A transformer equipped with KMCTA is referred to as a KMCT transformer. KMCTA ensures the causality

of time (in the time dimension, KMTA is bidirectional, while KMCTA is unidirectional). In addition, KMCTA always has a full-rank attention matrix, whereas KMTA and joint attention can not guarantee this property. The proof is detailed in *Appendix* A. Some studies Bhojanapalli et al. (2020); Han et al. (2023) indicate that it is important to avoid the low-rank bottleneck to improve the representation power of transformer architecture. This problem may become more knotty when transitioning images to videos.

### 3.3 INTERPRETIVE PROMPT

To address the issue 2) in Sec. 1, we prompt LLMs to generate interpretive texts that align the text semantics of video-text pairs with the rich text semantics in VLM's pre-training dataset. This design aids the model in understanding abstract verbs. This approach, termed interpretive prompt, designed from the following aspects: 1) Action decomposition: Providing detailed descriptions of actions by decomposing complex

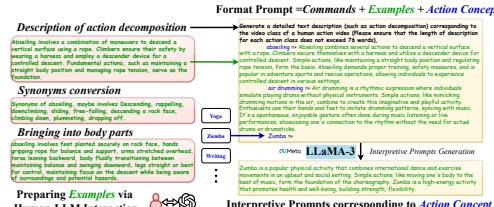

Figure 4: The Interpretive Prompt scheme.

actions into simpler, more basic ones. This clarifies the include relationship between complex and basic actions, and help distinguish similar actions, as not all action concepts are of equal status. For example, the action 'playing basketball' may consist of 'running', 'jumping', and 'shooting'. 'Running', 'jumping', and 'hand movements' are considered as basic actions, while 'playing basketball' is a complex action. Similarly, 'dribbling' may also involves 'running', and 'hand movements', besides, it is also a subset of 'playing basketball'. Action decomposition enhances the separability of action concepts in semantic space, helping models understand the relationships between actions. 2) Synonym conversion: Generating synonyms for verbs and phrases that convey the same core concept but with more varied expressions. This improves zero-shot generalization as an action concept may have multiple expressions. When encountering similar action descriptions in the unseen domain, the model demonstrates stronger generalization robustness. 3) Involving body parts: Describing actions based on possible body parts involved, helping the model to localize the region where action occurs.

We leverage ChatGPT OpenAI (2024) and LLaMA-3 Abhimanyu Dubey et al. (2024) to automatically generate action interpretations. Initially, we ask the knowledgeable ChatGPT OpenAI (2024) to provide several examples of text descriptions that align with our expect (based on the aforementioned aspects). Subsequently, we provide the format prompt to LLaMA-3 Abhimanyu Dubey et al. (2024) for text completion, as illustrated in Fig. 4. For example, "*Generate a detailed text description corresponding to the video class...*" –**Command**, accompanied by a few examples like, "*abseiling → Abseiling combines several actions...*" –**Examples**, and the action concept that LLM needs to interpret –**Action Concept**. We feed the format prompt to LLaMA-3 multiple times to obtain diverse interpretive prompts of actions. For a given action category, the texts used during training and inference include the original verbs or phrases, template filling prefixes or suffixes, and interpretive prompts. Our interpretive prompts provide texts at the sentence, phrase, and word-levels, enhancing the flexibility of text usage during the inference. More technique details refer to *Appendix* E.

### 3.4 TRAINING AND INFERENCE

In each training step, a batch of $B$ videos is sampled, and $M$ represents the total number of text descriptions of each verb or phrase. There are a total of $K$ action categories. The training loss is formulated as follows:

$$L = -\frac{1}{N} \sum_{m}^{\text{sub}\{M\}} \sum_{i}^{B} \log \frac{\exp(\text{sim}(\mathbf{v}^i, \mathbf{c}_m^i)/\tau)}{\sum_{k}^{K} \exp(\text{sim}(\mathbf{v}^i, \mathbf{c}_m^k)/\tau)}, \quad (9)$$

where, $\tau$ is the temperature parameter, $\text{sub}\{M\}$ indicates that we sample a subset of all text descriptions for a single training step. $\mathbf{v}^i$ represents the video belonging to the $i$-th category. $\mathbf{c}_m^i$ denotes the descriptions corresponding to the $i$-th category. During inference, we take the sum of the similarities between each $\mathbf{v}^i$ and all $\mathbf{c}_m^i$ as the final similarity score $S$:

$$S = \sum_{m}^{M} \log \frac{\exp(\text{sim}(\mathbf{v}^i, \mathbf{c}_m^i)/\tau)}{\sum_{k}^{K} \exp(\text{sim}(\mathbf{v}^i, \mathbf{c}_m^k)/\tau)}. \quad (10)$$

Table 1: Comparison with state-of-the-art on Kinetics-400. * indicates pretraining with a video-text collection. The FLOPs per view of each method is reported. Red represents optimal performance. **Bold** represents optimal performance among CLIP-based methods at the same model scale.

| Method | Pretrain | Frames | Top-1 | Top-5 | Views | FLOPs(G) |
|---|---|---|---|---|---|---|
| **Without Language** | | | | | | |
| Uniformer-B Li et al. (2022a) | IN-1k | 32 | 83.0 | 95.4 | 4×3 | 259 |
| TimeSformer-L Bertasius et al. (2021) | IN-21k | 96 | 80.7 | 94.7 | 1×3 | 2380 |
| Mformer-HR Patrick et al. (2021) | IN-21k | 16 | 81.1 | 95.2 | 10×3 | 959 |
| ATA Zhao et al. (2022) | IN-21k | 32 | 81.9 | 95.5 | 4×3 | 793 |
| Swin-L (384 ↑) Liu et al. (2022) | IN-21k | 32 | 84.9 | 96.7 | 10×5 | 2107 |
| MViTv2-L (312 ↑) Li et al. | IN-21k | 40 | 86.1 | 97.0 | 5×5 | 2828 |
| ViViT-H/14x2 Arnab et al. (2021) | JFT-300M | 32 | 84.9 | 95.8 | 4×3 | 8316 |
| TokenLearner-L/10 Ryoo et al. (2021) | JFT-300M | - | 85.4 | 96.3 | 4×3 | 4076 |
| CoVeR Zhang et al. (2021) | JFT-3B | - | 87.2 | - | 1×3 | - |
| **With Language** | | | | | | |
| MTV-H Yan et al. (2022) | WTS* | 32 | 89.1 | 98.2 | 4×3 | 3705 |
| X-CLIP-B/32 Ni et al. (2022) | CLIP-400M | 8 | 80.1 | 94.8 | 4×3 | 26 |
| X-CLIP-B/32 Ni et al. (2022) | CLIP-400M | 16 | 81.0 | 95.1 | 4×3 | 49 |
| ILA-B/32 Tu et al. (2023) | CLIP-400M | 8 | 80.6 | 94.9 | 4×3 | 40 |
| ILA-B/32 Tu et al. (2023) | CLIP-400M | 16 | 81.8 | 95.4 | 4×3 | 75 |
| **CLAVER-B/32 (KMT)** | CLIP-400M | 8 | **81.5** | **95.5** | 4×3 | 33 |
| **CLAVER-B/32 (KMT)** | CLIP-400M | 16 | **82.4** | **95.9** | 4×3 | 64 |
| **CLAVER-B/32 (KMCT)** | CLIP-400M | 8 | 81.4 | **95.5** | 4×3 | 33 |
| **CLAVER-B/32 (KMCT)** | CLIP-400M | 16 | **82.6** | **95.9** | 4×3 | 64 |
| Action-CLIP-B/16 Wang et al. (2021) | CLIP-400M | 16 | 82.6 | 96.2 | 10×3 | - |
| A6 Ju et al. (2022) | CLIP-400M | 16 | 76.9 | 93.5 | - | - |
| X-CLIP-B/16 Ni et al. (2022) | CLIP-400M | 8 | 83.1 | 96.5 | 4×3 | 92 |
| X-CLIP-B/16 Ni et al. (2022) | CLIP-400M | 16 | 84.3 | 96.8 | 4×3 | 183 |
| EVL-B/16 Lin et al. (2022) | CLIP-400M | 8 | 82.9 | - | - | 444 |
| EVL-B/16 Lin et al. (2022) | CLIP-400M | 16 | 83.6 | - | - | 888 |
| ViFiCLIP-B/16 Rasheed et al. (2023) | CLIP-400M | 16 | 83.9 | 96.3 | 4×3 | 281 |
| ASU-B/16 Chen et al. (2023) | CLIP-400M | 8 | 84.1 | 96.3 | 4×3 | 146 |
| ASU-B/16 Chen et al. (2023) | CLIP-400M | 16 | 84.8 | 96.7 | 4×3 | 288 |
| ILA-B/16 Tu et al. (2023) | CLIP-400M | 8 | 83.4 | 96.3 | 4×3 | 150 |
| ILA-B/16 Tu et al. (2023) | CLIP-400M | 16 | 85.0 | 97.0 | 4×3 | 302 |
| ALT-B/16 Chen et al. (2024b) | CLIP-400M | 16 | 85.5 | 96.7 | 3×1 | 1308 |
| **CLAVER-B/16 (KMT)** | CLIP-400M | 8 | **84.3** | 96.3 | 4×3 | 122 |
| **CLAVER-B/16 (KMT)** | CLIP-400M | 16 | 85.9 | **97.3** | 4×3 | 241 |
| **CLAVER-B/16 (KMCT)** | CLIP-400M | 8 | 84.1 | 96.2 | 4×3 | 122 |
| **CLAVER-B/16 (KMCT)** | CLIP-400M | 16 | **86.0** | 97.2 | 4×3 | 241 |
| X-CLIP-L/14 Ni et al. (2022) | CLIP-400M | 8 | 87.0 | 97.7 | 4×3 | 420 |
| X-CLIP-L/14 (336↑) Ni et al. (2022) | CLIP-400M | 16 | 87.6 | 97.5 | 4×3 | 1870 |
| EVL-L/14 Lin et al. (2022) | CLIP-400M | 8 | 86.3 | - | - | 2022 |
| EVL-L/14 (336↑) Lin et al. (2022) | CLIP-400M | 32 | 87.7 | - | - | 18196 |
| ASU-L/14 Chen et al. (2023) | CLIP-400M | 8 | 87.8 | 97.8 | 4×3 | 660 |
| ASU-L/14 (336↑) Chen et al. (2023) | CLIP-400M | 16 | 88.3 | 98.0 | 4×3 | 3084 |
| ILA-L/14 Tu et al. (2023) | CLIP-400M | 8 | 87.6 | 97.8 | 4×3 | 647 |
| ILA-L/14 (336↑) Tu et al. (2023) | CLIP-400M | 16 | 88.1 | 97.8 | 4×3 | 3130 |
| ALT-B/14 Chen et al. (2024b) | CLIP-400M | 16 | 87.8 | 97.7 | 3×1 | 4947 |
| **CLAVER-L/14 (KMT)** | CLIP-400M | 8 | **88.1** | 97.7 | 4×3 | 558 |
| **CLAVER-L/14(336↑) (KMT)** | CLIP-400M | 16 | 88.8 | **98.1** | 4×3 | 2488 |
| **CLAVER-L/14 (KMCT)** | CLIP-400M | 8 | 87.9 | 97.7 | 4×3 | 558 |
| **CLAVER-L/14(336↑) (KMCT)** | CLIP-400M | 16 | **88.9** | 98.0 | 4×3 | 2488 |

# 4 EXPERIMENTS

## 4.1 IMPLEMENTATION DETAILS

**Architectures and hyperparameters.** We employ CLIP-B/32, CLIP-B/16, CLIP-L/14 as our backbones and derive corresponding variants: CLAVER-B/32, CLAVER-B/16, CLAVER-L/14, respectively. The frame length settings include 8 and 16. For all CLAVER variants, the number of layers in the KMT/KMCT transformer is equal to one-third of the number of layers in the image encoder. For example, if the image encoder has 12 or 24 layers, then the KMT/KMCT transformer employs 4 or 8 layers. The detailed hyperparameter settings are provided in *Appendix* D.

**Datasets and metrics.** We evaluate the performance of our method on four benchmarks: Kinetics-400 Kay et al. (2017), Kientics-600 Carreira et al. (2018), UCF-101 Soomro et al. (2012), HMDB-51 Kuehne et al. (2011). We report the Top-1 and Top-5 accuracy as evaluation metrics.

## 4.2 COMPARISON RESULTS

**Fully-supervised Experiments.** We conducted fully supervised experiments on Kinetics-400 and Kinetics-600, respectively. In Tab. 1, we employ three variant models, CLAVER-B/32, CLAVER-B/16, and CLAVER-L/14, and sample 8 or 16 frames ($8f$,$16f$) with a sparse sampling for each model, employ KMT/KMCT, respectively. CLAVER-B/16$_{8f}$ (KMT/KMCT) and CLAVER-B/16$_{16f}$ (KMT/KMCT) surpass several methods Bertasius et al. (2021); Patrick et al. (2021); Zhao et al. (2022); Li et al. (2022a) pretrained on ImageNet-1k/21k Deng et al. (2009) with shorter frames.

Table 2: Comparison with state-of-the-art on Kinetics-600.

| Method | Pretrain | Frames | Top-1 | Top-5 | Views |
|---|---|---|---|---|---|
| MViT-B-24 Fan et al. (2021) | - | 32 | 83.8 | 96.3 | 5×1 |
| Swin-L (384 ↑) Liu et al. (2022) | JFT-300M | 32 | 85.8 | 96.5 | 4×3 |
| ViViT-H/14x2 Arnab et al. (2021) | JFT-300M | 32 | 85.8 | 96.5 | 4×3 |
| TokenLearner-L/10 Ryoo et al. (2021) | JFT-300M | - | 86.3 | 97.0 | 4×3 |
| CoVeR Zhang et al. (2021) | JFT-3B | 32 | 87.9 | - | 4×3 |
| MTV-H Yan et al. (2022) | WTS* | 32 | 89.6 | 98.3 | 4×3 |
| Florence (384 ↑) Yuan et al. (2021) | FLD-900M | - | 87.8 | - | 1×3 |
| X-CLIP-B/16 Ni et al. (2022) | CLIP-400M | 8 | 85.3 | 97.1 | 4×3 |
| ASU-B/16 Chen et al. (2023) | CLIP-400M | 8 | 85.7 | - | 4×3 |
| **CLAVER-B/16 (KMT)** | CLIP-400M | 8 | 85.9 | 97.3 | 4×3 |

Table 3: Few-shot performance on HMDB-51 and UCF-101.

| Method | HMDB-51 | | | | UCF-101 | | | |
|---|---|---|---|---|---|---|---|---|
| | K=2 | K=4 | K=8 | K=16 | K=2 | K=4 | K=8 | K=16 |
| TSM Lin et al. (2019) | 17.5 | 20.9 | 18.4 | 31.0 | 25.3 | 47.0 | 64.4 | 61.0 |
| TimeSformer Bertasius et al. (2021) | 19.6 | 40.6 | 49.4 | 55.4 | 48.5 | 75.6 | 83.7 | 89.4 |
| Swin-B Liu et al. (2022) | 20.9 | 41.3 | 47.9 | 56.1 | 53.3 | 74.1 | 85.8 | 88.7 |
| Action-CLIP Wang et al. (2021) | 55.0 | 56.0 | 58.0 | - | 80.0 | 85.0 | 89.0 | - |
| X-CLIP Ni et al. (2022) | 53.0 | 57.3 | 62.8 | 64.0 | 76.4 | 83.4 | 88.3 | 91.4 |
| X-Florence Ni et al. (2022) | 51.6 | 57.8 | 64.1 | 64.2 | 84.0 | 88.5 | 92.5 | 94.8 |
| MAXI Lin et al. (2023) | 58.0 | 60.1 | 65.0 | 66.5 | 86.8 | 89.3 | 92.4 | 93.5 |
| ASU Chen et al. (2023) | **60.1** | 63.8 | 67.2 | 70.8 | **91.4** | **94.6** | 96.0 | 97.2 |
| **CLAVER (KMT)** | 58.6 | **63.9** | **68.0** | **72.5** | 89.7 | 92.9 | **96.1** | **98.0** |

Table 4: Zero-shot on HMDB-51 and UCF-101.

| Method | HMDB-51 | UCF-101 |
|---|---|---|
| MTE Xu et al. (2016) | 19.7 ± 1.6 | 15.8 ± 1.3 |
| ASR Wang & Chen (2017) | 21.8 ± 0.9 | 24.4 ± 1.0 |
| ZSECOC Qin et al. (2017) | 22.6 ± 1.2 | 15.1 ± 1.7 |
| UR Zhu et al. (2018) | 24.4 ± 1.6 | 17.5 ± 1.6 |
| TS-GCN Gao et al. (2019) | 23.2 ± 3.0 | 34.2 ± 3.1 |
| E2E Brattoli et al. (2020) | 32.7 | 48 |
| ER-ZSAR Chen & Huang (2021) | 35.3 ± 4.6 | 51.8 ± 2.9 |
| Action-CLIP Wang et al. (2021) | 40.8 ± 5.4 | 58.3 ± 3.4 |
| X-CLIP Ni et al. (2022) | 44.6 ± 5.2 | 72.0 ± 2.3 |
| ASU Chen et al. (2023) | 48.1 ± 2.8 | 75.0 ± 2.3 |
| MAXI Lin et al. (2023) | 51.2 ± 1.1 | 75.2 ± 0.9 |
| OST Chen et al. (2024a) | 52.9 ± 0.9 | 75.3 ± 2.1 |
| **CLAVER (KMT)** | **54.1 ± 2.4** | **78.6 ± 1.7** |

Table 5: Zero-shot on Kinetics-600.

| Method | Top-1 | Top-5 |
|---|---|---|
| DEVISE Frome et al. (2013) | 23.8 ± 0.3 | 51.0 ± 0.6 |
| ALE Akata et al. (2015a) | 23.4 ± 0.8 | 50.3 ± 1.4 |
| SJE Akata et al. (2015b) | 22.3 ± 0.6 | 48.2 ± 0.4 |
| ESZSL Romera-Paredes & Torr (2015) | 22.9 ± 1.2 | 48.3 ± 0.8 |
| DEM Zhang et al. (2017) | 23.6 ± 0.7 | 49.5 ± 0.4 |
| GCN Ghosh et al. (2020) | 22.3 ± 0.6 | 49.7 ± 0.6 |
| ER-ZSAR Chen & Huang (2021) | 42.1 ± 1.4 | 73.1 ± 0.3 |
| X-CLIP Ni et al. (2022) | 65.2 ± 0.4 | 86.1 ± 0.8 |
| ASU Chen et al. (2023) | 67.6 ± 0.2 | 87.2 ± 0.3 |
| MAXI Lin et al. (2023) | 70.9 ± 1.2 | 92.1 ± 0.5 |
| OST Chen et al. (2024a) | 70.5 ± 0.7 | 92.1 ± 0.3 |
| **CLAVER (KMT)** | **73.8 ± 0.6** | **93.1 ± 0.6** |

CLAVER-B/$16_{16f}$ (KMT) outperforms Swin-L (384↑) Liu et al. (2022) by 1.0%, and is slightly lower than MViTv2-L (312↑) Li et al., as lower resolution, shorter frames and fewer views. CLAVER-B/$14_{8f}$ (KMT/KMCT) outperforms some methods Arnab et al. (2021); Ryoo et al. (2021); Zhang et al. (2021) pretrained on JFT-300M/JFT-3B. CLAVER-B/$14_{16f}$ (KMT) outperforms CoVeR Zhang et al. (2021) by 1.6%, however, inferior to MTV-H Yan et al. (2022), as it utilizes WTS*, which contains 70M video-text pairs with about 17B images, much larger than CLIP-400M. Compared to those approaches based on CLIP Radford et al. (2021), under configuration ViT-B/$32_{8f}$ and ViT-B/$32_{16f}$, CLAVER (KMT/KMCT) surpasses X-CLIP Ni et al. (2022) and ILA Tu et al. (2023). Under configuration ViT-B/$16_{8f,16f}$ and ViT-L/$14_{8f,16f}$, CLAVER (KMT/KMCT) exceeds most methods under the same configuration. In Tab. 2, CLAVER-B/$16_{8f}$ (KMT) achieves higher performance compared to ViViT-H/14x2 Arnab et al. (2021), MViT-B-24 Fan et al. (2021). Our method has lower performance than these methods Ryoo et al. (2021); Zhang et al. (2021); Yan et al. (2022); Yuan et al. (2021); Liu et al. (2022), as they use longer frames, or more data, or higher resolutions. In addition, CLAVER-B/$16_{8f}$ (KMT/KMCT) outperforms ASU-B/$16_{8f}$ Chen et al. (2023) and X-CLIP-B/$16_{8f}$ Ni et al. (2022) at the same scale.

**Few-shot Experiments.** We pretrain CLAVER-B/$16_{32f}$ (KMT) ($32f$ means with 32 frames) on Kinetics-400, and then perform few-shot transfer of 2, 4, 8, 16 samples on HMDB-51 and UCF-101. Tab. 3 depicts the results of few-shot experiments. CLAVER is comparable wtih ASU Chen et al. (2023), and consistently surpasses the X-CLIP Ni et al. (2022) and MAXI Lin et al. (2023) across all K ranges. Additionally, CLAVER significantly outperforms other previous methods like Lin et al. (2019); Bertasius et al. (2021); Liu et al. (2022). More details about the evaluation protocols are provided in the *Appendix* D.

**Zero-shot Experiments.** We also pretrain CLAVER-B/$16_{32f}$ (KMT) on Kinetics400 for zero-shot. As shown in Tab. 4, on HMDB-51 Kuehne et al. (2011) and UCF-101 Soomro et al. (2012) benchmarks, CLAVER surpasses OST Chen et al. (2024a) , ASU Chen et al. (2023) and X-CLIP Ni et al. (2022) under the same configuration, and far outperforms other previous methods. Additionally, in Tab. 5, on Kinetics600 Carreira et al. (2018) benchmark, CLAVER outperforms OST Chen et al. (2024a), MAXI Lin et al. (2023) and all other methods. More details about the evaluation protocols are provided in the *Appendix* D.

## 4.3 ABLATION STUDY

**Components ablation studies.** We performed ablation studies to evaluate the effects of each component under the CLAVER-B/$32_{8f}$ (KMT) configuration. The results are shown in Tab. 6. Our baseline, denoted as CLIP-Mean, implements temporal mean pooling for CLIP. By equipping the CLIP with a KMT transformer at 1/3 scale, the Top-1 accuracy increases by 3.5%. We only introduce interpretive prompt, the performance increases by 1%. When we further incorporate both of them, CLAVER surpasses the CLIP-Mean by 4.1%. In addition, we test the effect of the number of KMT transformer layers. With only one layer, the performance improvement is minimal. Increasing

Table 6: Ablation of each component.

| Components | Top-1 (%) |
|---|---|
| Baseline (CLIP-Mean) | 77.4 |
| Baseline + KMTA 1 layer | 78.3 |
| Baseline + KMTA 1/6 scale | 79.6 |
| Baseline + KMTA 1/3 scale | 80.9 |
| Baseline + Interpretive Prompt | 78.4 |
| Baseline + KMTA 1/3 scale + Interpretive Prompt | **81.5** |

Table 7: Comparison of temporal attentions.

| Temporal Modeling | Top-1 (%) | Top-5 (%) |
|---|---|---|
| Baseline (Mean Pooling) | 78.4 | 94.3 |
| Class-Token-Only | 78.9 | 94.3 |
| Pipeline Temporal Attention | 79.4 | 94.4 |
| Joint Attention | 80.1 | 94.9 |
| Kronecker Mask Causal Temporal Attention | 81.4 | **95.5** |
| Kronecker Mask Temporal Attention | **81.5** | **95.5** |

Table 8: The impact of patch size and frame length on joint attention (JA), KMTA and KMCTA. Conducting on HMDB-51 and UCF-101.

| HMDB-51, UCF-101 | patch size = 32 (ViT-B/32) | | | patch size = 16 (ViT-B/16) | | |
|---|---|---|---|---|---|---|
| | JA (%) | KMTA (%) | KMCTA (%) | JA (%) | KMTA (%) | KMCTA (%) |
| frame length = 8 | 67.9, 93.3 | 68.8, 94.1 | 68.7, 93.9 | 72.2, 96.2 | 73.2, 96.3 | 72.2, 96.1 |
| frame length = 16 | 68.0, 93.6 | 69.1, 94.2 | 69.4, 94.8 | 72.1, 96.3 | 72.3, 96.6 | 72.8, 96.4 |

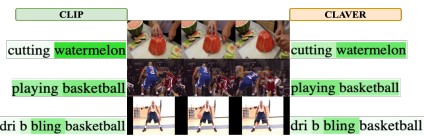
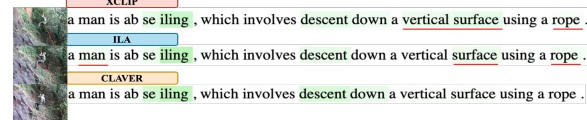

Figure 5: Word importance of CLIP, X-CLIP, ILA and CLAVER. Darker color, higher importance.

the number of layers to 1/6 scale results in further performance gains, and at 1/3 scale, we observe significant improvement.

**Comparison of different temporal attentions and low-rank bottleneck issue.** Tab. 7 compares the performance of different temporal modeling methods mentioned in Sec. 3.2. We find that class-token-only temporal attention and pipeline temporal attention have inferior performance. KMTA and KMCTA outperform joint attention. The top-1 (%) of KMTA slightly surpasses KMCTA by 0.1%. Then, we conduct further experiments on HMDB-51 and UCF101 that observing the impact of the low-rank bottleneck issue on them. We increasing the length of token sequences by reducing patch size or increasing frame length, and observe the effects on them, as shown in Tab. 8. When reducing the patch size, however, we do not observe the low-rank bottleneck. Joint attention, KMTA and KMCTA achieve better performance due to more fine-grained features as the smaller patch size results in each token representing smaller local region. In contrast, when we increase the frame length, KMCTA's performance is optimal in most configurations, and only the performance of KM-CTA can steadily improve. The performance improvement of joint attention and KMTA is limited. Meanwhile, in Tab. 1, with the increase of frame length under the same backbone, the performance increase of KMCTA is also greater than that of KMTA. This indicates that KMCTA has a more significant advantage with longer frame length.

## 5 VISUALIZATION AND ANALYSIS

We employ Chefer et al. (2021) for bi-modal visualization and show the explainability of visual-textual attentions.

**Word importance**. We visualize word importance for CLIP, X-CLIP, ILA and CLAVER in Fig. 5. Observations indicate that CLIP tends to focus on nouns, whereas CLAVER prefers to verbs. Compared to previous works, CLAVER is more inclined towards verb concept, while X-CLIP Ni et al. (2022) and ILA Tu et al. (2023) show a slight inclination towards nouns. These findings indicate the effectiveness of interpretive prompts for nouns concept to verbs concept transition.

**Spatiotemporal homogenization study**. We define spatiotemporal homogenization as a phenomenon where random token shuffling has little impact on the semantics of the visual branch, which is illogical. It aims to illustrate the interpretability behind the performance of spatiotemporal modeling. In Fig. 6 (**Upper Left**), we illustrate the token shuffling. We denote token shuffling before adding time embedding as PreTE shuffling, token shuffling after adding time embedding as PostTE shuffling, and None represents no shuffle. Fig. 6 shows changes in word importance and similarity following both PreTE and PostTE shuffling. For joint attention, we observe that PreTE shuffling marginally affects similarity and word importance, while PostTE shuffling does not affect

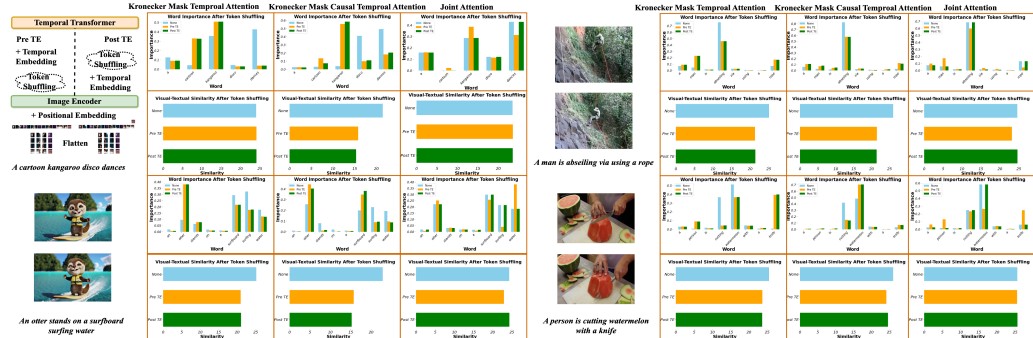

Figure 6: Spatiotemporal Homogenization. (**Upper Left**) Token shuffling. (**Vertical bar chart**) Word importance refers to the degree of correlation between each word in a sentence description and the semantics of the video content, while (**Horizontal bar chart**) Similarity refers to the similarity between visual and textual representations.

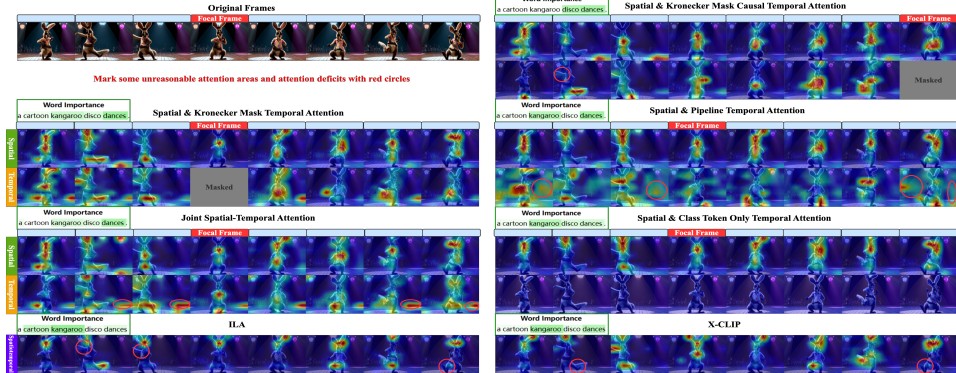

Figure 7: Visualizing spatiotemporal attention maps of different spatio-temporal modeling.

similarity score and word importance. Regarding KMTA, both PreTE and PostTE shuffling lead to changes in the similarity and word importance. Meanwhile, KMCTA is profoundly affected by both PreTE and PostTE shuffling, which result in lower similarity and disturbance of word importance. This phenomenon suggest that KMTA and KMCTA possess varying degrees ability in mitigating spatiotemporal homogenization and the Kronecker mask serves as a natural inductive bias for spatiotemporal structural heterogeneity. It also demonstrates that equipping learnable position/time encoding is not insufficient to alleviate spatiotemporal homogenization. Please refer to *Appendix* B for more and explanations and visualizations.

**Visualization of spatiotemporal attention map**. Fig. 7 presents various spatiotemporal attention maps. X-CLIP Ni et al. (2022) and ILA Tu et al. (2023) adopt factorized attention structures resulting in intertwined spatiotemporal attention maps. However, CLAVER employs a factorized encoder structure, allowing for the separation of spatial and temporal attention maps. Notably, class-token-only temporal attention discards other time tubes so that its temporal attention maps appear vacant. KMTA and KMCT have more reasonable spatial (focuses on the executor of the action) and temporal (focuses on areas where the action occurs) attention maps, while ILA and X-CLIP ignore some action dense areas. Moreover, joint attention and pipeline temporal attention will be attracted by some irrelevant backgrounds.

## 6 CONCLUSION

In this work, we present the **C**ontrastive **L**anguage-**A**ction **V**ideo Learn**er** (**CLAVER**) to shift from the alignment of visual objects and nouns in CLIP to the alignment of action behaviors and verbs. We propose Kronecker mask temporal attention and Kronecker mask causal temporal attention for temporal modeling. Interpretive prompts are employed to transition the focus on nouns to verbs. Extensive experiments under different evaluation settings demonstrate the effectiveness of our method.

## 7 ACKNOWLEDGEMENT

The numerical calculations in this paper have been done on the supercomputing system in the Supercomputing Center of University of Science and Technology of China. This work was supported by the National Science Foundation of China, under Grant No. 62171425. This work was supported by National Natural Science Foundation of China under Grant 62306061. This work was supported by Guangdong Basic and Applied Basic Research Foundation (Grant No. 2023A1515140037). This work was supported by Guangdong Research Team for Communication and Sensing Integrated with Intelligent Computing (Project No. 2024KCXTD047).

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

# 8 APPENDIX

**Roadmap**

In the Appendix, we present proof in A, Spatiotemporal homogenization study details in B, some additional experiment results in C, experiment setting details in D, interpretive prompt technique details in E, analysis of synthetic video testing in F, dataset details in G, and more visualization of spatiotemporal attention map in H.

## A PROOF

We conclude that there is a low-rank bottleneck problem for joint attention and KMTA when facing longer token sequence lengths ($d \ll n$), while KMCTA always guarantees the property of full rank under our assumption. Proofs as follow:

### A.1 PROOF OF JOINT ATTENTION CANNOT GUARANTEE FULL RANK

Based on the Representation Theorem of Bhojanapalli et al. Bhojanapalli et al. (2020):

**Theorem 1** (Representation Theorem Bhojanapalli et al. (2020)). *If $d_q = d_k = d \geq n$, then given any full row rank matrix $\mathbf{X} \in \mathcal{R}^{n \times d}$ and an arbitrary $n \times n$ positive row stochastic matrix $\mathbf{P}$, subject to the sum of each row of $\mathbf{P}$ is equal to 1, i.e., $\mathbf{P1} = \mathbf{1}$, there always exists $d \times d$ projection matrices $\mathbf{W}_q$ and $\mathbf{W}_k$ such that*

$$\text{Softmax}[\frac{(\mathbf{XW}_q)(\mathbf{XW}_k)^T}{\sqrt{d_k}}] = \mathbf{P} \tag{11}$$

*If $d_q = d_k = d < n$, there exist $\mathbf{X}$ and $\mathbf{P}$ such that Eqn. 11 does not hold for all $\mathbf{W}_q$ and $\mathbf{W}_k$.*

The proof process of **Theorem 1** are referred from Bhojanapalli et al. Bhojanapalli et al. (2020). By the way, in Bhojanapalli et al. Bhojanapalli et al. (2020), they set $\mathbf{X} \in \mathcal{R}^{d \times n}$. However, for the convenience of our subsequent proof, we transpose the $n$ and $d$ dimensions, $\mathbf{X} \in \mathcal{R}^{n \times d}$.

For $d \geq n$ case. Since $\mathbf{X}$ exhibits full row rank, there exists a right pseudo inverse $\mathbf{X}^{\dagger} = \mathbf{X}^T(\mathbf{XX}^T)^{-1} \in \mathcal{R}^{d \times n}$ such that $\mathbf{XX}^{\dagger} = \mathbf{I}_n$. Let $\mathbf{W}_k = \mathbf{X}^{\dagger}\tilde{\mathbf{W}}_k$ and $\mathbf{W}_q = \mathbf{X}^{\dagger}\tilde{\mathbf{W}}_q$. Then

$$(\mathbf{XW}_q)(\mathbf{XW}_k)^T = \mathbf{XW}_q\mathbf{W}_k^T\mathbf{X}^T \tag{12}$$

$$= \mathbf{XX}^{\dagger}\tilde{\mathbf{W}}_q\tilde{\mathbf{W}}_k^T(\mathbf{X}^{\dagger})^T\mathbf{X}^T \tag{13}$$

$$= \mathbf{XX}^{\dagger}\tilde{\mathbf{W}}_q\tilde{\mathbf{W}}_k^T(\mathbf{XX}^{\dagger})^T \tag{14}$$

$$= \mathbf{I}_n\tilde{\mathbf{W}}_q\tilde{\mathbf{W}}_k^T\mathbf{I}_n^T \tag{15}$$

$$= \tilde{\mathbf{W}}_q\tilde{\mathbf{W}}_k^T = \tilde{\mathbf{W}}_{qk} \tag{16}$$

According Eqn. 11, we obtain that

$$\text{Softmax}[\frac{(\mathbf{XW}_q)(\mathbf{XW}_k)^T}{\sqrt{d_k}}] = \text{Softmax}[\frac{\tilde{\mathbf{W}}_{qk}}{\sqrt{d_k}}] \tag{17}$$

$$= \mathbf{D}_{\tilde{\mathbf{W}}_{qk}}^{-1}\exp(\frac{\tilde{\mathbf{W}}_{qk}}{\sqrt{d_k}}) \tag{18}$$

where $\mathbf{D}_{\tilde{\mathbf{W}}_{qk}}$ is a $n \times n$ diagonal matrix such that

$$(\mathbf{D}_{\tilde{\mathbf{W}}_{qk}})_{ii} = \sum_{j=1}^{n}\exp(\frac{(\tilde{\mathbf{W}}_{qk})_{ji}}{\sqrt{d_k}}) \tag{19}$$

$$= (\exp(\frac{\tilde{\mathbf{W}}_{qk}}{\sqrt{d_k}})\mathbf{1})_i \tag{20}$$

We can establish the desired result by showing that there always exists a $\tilde{\mathbf{W}}_{qk}$ that satisfies the following fixed point equation:

$$\mathbf{D}_{\tilde{\mathbf{W}}_{qk}}^{-1} \exp(\frac{\tilde{\mathbf{W}}_{qk}}{\sqrt{d_k}}) = \mathbf{P} \tag{21}$$

Given $\mathbf{P}$, to construct such a $\tilde{\mathbf{W}}_{qk}$, we can pick an random positive diagonal matrix $\mathbf{D}_0$:

$$\tilde{\mathbf{W}}_{qk} = \sqrt{d_k} \cdot \log(\mathbf{D}_0 \mathbf{P}) \tag{22}$$

Since $\mathbf{P}$ is a positive matrix, and $\mathbf{D}_0$ is a positive diagonal matrix, such a $\tilde{\mathbf{W}}_{qk}$ always exists. According to Eqn. 19 we can conclude

$$\mathbf{D}_{\tilde{\mathbf{W}}_{qk}} = \text{Diag}(\exp(\frac{\tilde{\mathbf{W}}_{qk}}{\sqrt{d_k}})\mathbf{1}) \tag{23}$$

$$= \text{Diag}(\mathbf{D}_0 \mathbf{P})) = \text{Diag}(\mathbf{D}_0 \mathbf{1})) = \mathbf{D}_0 \tag{24}$$

which indicates that $\exp(\frac{\tilde{\mathbf{W}}_{qk}}{\sqrt{d_k}}) = \mathbf{D}_0 \mathbf{P} = \mathbf{D}_{\tilde{\mathbf{W}}_{qk}} \mathbf{P}$. This completes the proof of $d \geq n$ case of **Theorem 1**.

Regarding $d < n$ case. Consider the special case of $d = 1$ and $n = 2$. There has $\mathbf{X} \in \mathcal{R}^{2 \times 1}$ and $\mathbf{W}_q, \mathbf{W}_k \in \mathcal{R}^{1 \times 1}$. Suppose $\mathbf{X} = \begin{bmatrix} 1 \\ 0 \end{bmatrix}$, so that:

$$\text{Softmax}[\frac{(\mathbf{X}\mathbf{W}_q)(\mathbf{X}\mathbf{W}_k)^T}{\sqrt{d_k}}] = \text{Softmax}[\frac{\begin{bmatrix} 1 \\ 0 \end{bmatrix} \mathbf{W}_q \mathbf{W}_k^T [1, 0]}{\sqrt{d_k}}] \tag{25}$$

$$= \text{Softmax}[\begin{bmatrix} \frac{\mathbf{W}_q \mathbf{W}_k^T}{\sqrt{d_k}} & 0 \\ 0 & 0 \end{bmatrix}] \tag{26}$$

This matrix clearly cannot be used to generate $\mathbf{P}$. Then we extend the above special case to general values of $n$ and $d$, $(d < n)$. Let $\mathbf{X} = [\mathbf{1}_d, \cdots, \mathbf{1}_d, \mathbf{0}_d]^T = [\mathbf{1}_{\text{mat}}, \mathbf{0}_d]^T \in \mathcal{R}^{n \times d}$, where $\mathbf{1}_d, \mathbf{0}_d \in \mathcal{R}^d$ denotes the all ones, zeros column vector, and $\mathbf{1}_{\text{mat}}$ denotes the $d \times (n-1)$ all ones matrix. Then

$$\text{Softmax}[\frac{(\mathbf{X}\mathbf{W}_q)(\mathbf{X}\mathbf{W}_k)^T}{\sqrt{d_k}}] = \text{Softmax}[\frac{[\mathbf{1}_{\text{mat}}, \mathbf{0}_d]^T \mathbf{W}_q \mathbf{W}_k^T [\mathbf{1}_{\text{mat}}, \mathbf{0}_d]}{\sqrt{d_k}}] \tag{27}$$

$$= \text{Softmax}[\begin{bmatrix} \mathbf{1}_{\text{mat}}^T \frac{\mathbf{W}_q \mathbf{W}_k^T}{\sqrt{d_k}} \mathbf{1}_{\text{mat}} & \mathbf{0}_{n-1} \\ \mathbf{0}_{n-1} & 0 \end{bmatrix}] \tag{28}$$

The basic idea of the proof remains consistency, and we can conclude the same conclusion. According to the **Theorem 1**, we can not ensure the attention matrix of joint attention to be full rank.

## A.2 PROOF OF KMCTA CAN GUARANTEE FULL RANK

**Assumption 1** The elements of $\frac{(\mathbf{X}\mathbf{W}_q)(\mathbf{X}\mathbf{W}_k)^T}{\sqrt{d_k}}$ does not exist negative infinity in normal cases, which means that 0 will not appear at any other position in the attention matrix $\tilde{\mathbf{A}} = \text{Softmax}[\frac{(\mathbf{X}\mathbf{W}_q)(\mathbf{X}\mathbf{W}_k)^T}{\sqrt{d_k}} + \tilde{\mathbf{M}}]$ except for the masked position.

The softmax operation can ensure that the elements at all other positions are greater than 0 except for the masked position. The attention matrix $\tilde{\mathbf{A}}$ of KMCTA is always a lower triangular matrix due to the presence of the $\tilde{\mathbf{M}}$ and the diagonal elements are always greater than 0, i.e., $\forall a_{ii} > 0$:

$$\tilde{\mathbf{A}} = \begin{bmatrix} a_{11} & 0 & 0 & \cdots & 0 \\ a_{21} & a_{22} & 0 & \cdots & 0 \\ a_{31} & a_{32} & a_{33} & \cdots & 0 \\ \vdots & \vdots & \vdots & \ddots & \vdots \\ a_{n1} & a_{n2} & a_{n3} & \cdots & a_{nn} \end{bmatrix} \tag{29}$$

The determinant of the $\tilde{\mathbf{A}}$, $|\tilde{\mathbf{A}}| = a_{11} a_{22} \cdots a_{nn} > 0 \neq 0$, because $\forall a_{ii} > 0$. Thus, the attention matrix $\tilde{\mathbf{A}}$ of KMCTA is always reversible, i.e. always be full rank.

### A.3 PROOF OF KMTA CANNOT GUARANTEE FULL RANK

According to the **Assumption 1**. We consider a special case, when $T=2$ and $L=2$, the attention matrix $\mathbf{A} = \text{Softmax}[\frac{(\mathbf{XW}_q)(\mathbf{XW}_k)^T}{\sqrt{d_k}} + \mathbf{M}]$ of KMTA can be formulated as:

$$
\begin{bmatrix}
a_{11} & 0 & a_{13} & a_{14} \\
0 & a_{22} & a_{23} & a_{24} \\
a_{31} & a_{32} & a_{33} & 0 \\
a_{41} & a_{42} & 0 & a_{44}
\end{bmatrix}
\rightarrow
\begin{bmatrix}
a_{11} & 0 & a_{13} & a_{14} \\
0 & a_{22} & a_{23} & a_{24} \\
0 & 0 & a_{33} - \frac{a_{13}}{a_{11}}a_{31} - \frac{a_{23}}{a_{22}}a_{32} & -\frac{a_{14}}{a_{11}}a_{31} - \frac{a_{24}}{a_{22}}a_{32} \\
0 & 0 & -\frac{a_{13}}{a_{11}}a_{41} - \frac{a_{23}}{a_{22}}a_{42} & a_{44} - \frac{a_{14}}{a_{11}}a_{41} - \frac{a_{24}}{a_{22}}a_{42}
\end{bmatrix}
\tag{30}
$$

where $a_{11} + a_{13} + a_{14} = 1, a_{22} + a_{23} + a_{24} = 1, a_{31} + a_{32} + a_{33} = 1, a_{41} + a_{42} + a_{44} = 1$. When $a_{33} - \frac{a_{13}}{a_{11}}a_{31} - \frac{a_{23}}{a_{22}}a_{32} = -\frac{a_{13}}{a_{11}}a_{41} - \frac{a_{23}}{a_{22}}a_{42}$, $\mathbf{A}$ can not always be full rank. For example:

$$
\mathbf{A} = \begin{bmatrix}
0.4 & 0 & 0.4 & 0.2 \\
0 & 0.4 & 0.4 & 0.2 \\
0.4 & 0.5 & 0.1 & 0 \\
0.4 & 0.4 & 0 & 0.2
\end{bmatrix}
\tag{31}
$$

For a video, $n = T \times L$, If we fix the number of attention heads, the dimension of tokens, the patch size, and the resolution of the input image, the only factor that can affect $n$ for a video is the frame length. When we pray for a longer frame length and an increase in the number of tokens to improve performance, we may face the problems pointed out in **Theorem 1**. Because increasing the frame length will further increase $n$, the expressive power of self attention may encounter bottlenecks. The low-rank bottleneck is vital, because it may lead to that many rows of the attention map are seriously homogenized. As the output of self-attention is the weighted sum of the same set of value vectors, the homogenization of attention weights inevitably leads to the resemblance among the aggregated features.

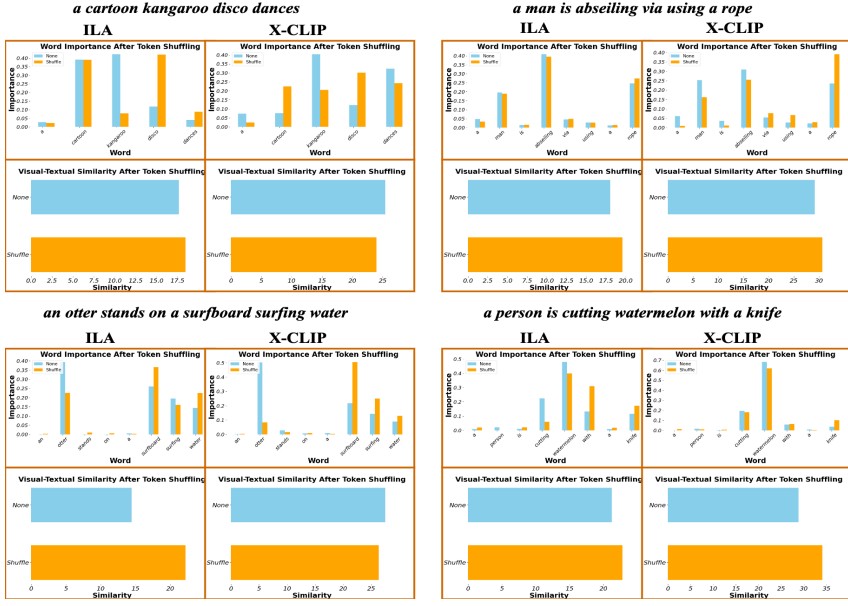

Figure 8: We visualize the impact of token shuffling on the ILA, XCLIP. In most cases, token shuffling has limited influence on their word importance. Surprisingly, in some figures, after token shuffling, the similarity actually increases, which is illogical. This indicates that the response of ILA and XCLIP to spatiotemporal position destruction is uncertain and has poor interpretability.

## B SPATIOTEMPORAL HOMOGENIZATION STUDY

Fig. 6 shows that equipping learnable position/time encoding with joint attention is not insufficient to alleviate spatiotemporal homogenization. For example, (Upper Left) 'a cartoon kangaroo disco

dances'. We can observe that token shuffling has a limited impact on word importance and similarity of joint attention, but it has a significant impact on the word importance of KMT and both similarity and word importance of KMCT. (Upper Right) 'a man is abseiling via using a rope'. (Lower Left) 'an otter stands on a surfboard surfing water'. (Lower Right) 'a person is cutting watermelon with a knife'. The key point is that classification accuracy does not necessarily indicate that these learnable position encoding or attention biases genuinely grasp the concept of spatiotemporal structure. Since the optimization goal of the network is usually to pursue performance, the learned positional encoding or attention bias serve to improve accuracy rather than understand the spatiotemporal concept. This is because their optimization is guided by gradients aimed at increasing accuracy, rather than by an objective function that tells them what spatiotemporal structure is. The Kronecker mask in KMTA and KMCTA pursues not only for good performance, but also for interpretability. The Kronecker mask act as a natural spatiotemporal heterogeneity inductive bias, KMTA/KMCTA exhibit better spatiotemporal complementarity with spatial attention. Additionally, in Fig. 8, we show that several tailored modules (ILA and XCLIP) also encounter spatiotemporal homogenization issues during our testing. Surprisingly, in some cases like 'a man is abseiling via using a rope,' 'an otter stands on a surfboard surfing water,' and 'a person is cutting watermelon with a knife,' the similarity actually increases after token shuffling, which is illogical.

Additionally, we evaluate the performance of joint attention, Kronecker mask temporal attention (KMTA) and Kronecker mask causal temporal attention (KMCTA) on K400 Kay et al. (2017) after introducing token shuffling. The results are shown in Tab. 9. The greater performance degradation observed in KMTA and KMCTA indicates that they are more sensitive to disturbances in the spatiotemporal structure, thus alleviating spatiotemporal homogenization.

Table 9: Performance of joint attention, KMTA and KMCTA while introducing token shuffling.

| Token Shuffling (Yes-Y, No-N) | Top-1 (%) |
|---|---|
| Joint Attention, N | 80.1 |
| Kronecker Mask Temporal Attention, N | 81.5 |
| Kronecker Mask Causal Temporal Attention, N | 81.4 |
| Joint Attention, Y | 58.9 (-21.2) |
| Kronecker Mask Temporal Attention, Y | 43.5 (-38.0) |
| Kronecker Mask Causal Temporal Attention, Y | 42.6 (-38.8) |

## C  ADDITIONAL EXPERIMENTS

**The effects of interpretive prompt.** In Tab. 10, we can observe that, directly use verbs and phrases achieve the lowest performance. When employing prefix and suffix prompt templates, the performance slightly improved. When introducing action decomposition interpretive prompts, the performance significantly improved. When adding synonym conversion interpretive prompts, the performance is further increased. Last, we involve body parts interpretive prompts that the performance reaches the best.

**Which branches should be finetuned?** We separately freeze the parameters of the pretrained image and text encoder. From Tab. 11, we conclude the following observations: 1) Freezing both image and text encoders, only tuning KMT transformer achieves the worst performance. 2). Finetuning only one of them will improve the performance. 3). Finetuning them simultaneously achieves the best performance.

**The Evaluation performance of zero- and few-shot experiments on Kinetics-600, HMDB-51, and UCF-101 with CLAVER-B/16 (KMCT) configuration.** The comparison of KMT and KMCT as shown in Tab. 12, 13. In the zero-shot scenario, the performance of KMCTA and KMTA has its own wins and losses, which is due to the lack of further fine-tuning, and the model's ability is relatively dependent on previous training. In the few-shot scenario, we can observe that in most cases, the performance of KMCTA is better, which further indicates that when the scale of training data is limited, the low-rank bottleneck is more significant.

Table 10: Ablation on interpretive prompt.

| Text Prompts | Top-1 (%) | Top-5 (%) |
|---|---|---|
| Noun and Phrase of Action | 77.4 | 93.5 |
| + Prefix-Suffix | 77.4 | 93.6 |
| + Action Decomposition Interpretive Prompt | 78.0 | 94.0 |
| + Synonym Conversion Interpretive Prompt | 78.2 | 94.2 |
| + Involving Body Parts Interpretive Prompt | **78.4** | **94.3** |

Table 11: Which branches should be finetuned.

| KMT transformer | Visual | Text | Top-1 (%) | Top-5 (%) |
|---|---|---|---|---|
| ✓ | ✗ | ✗ | 78.7 | 94.4 |
| ✓ | ✓ | ✗ | 81.0 | 95.1 |
| ✓ | ✗ | ✓ | 80.8 | 95.2 |
| ✓ | ✓ | ✓ | **81.5** | **95.5** |

Table 12: Comparison of KMT and KMCT on zero-shot settings.

| Zero-shot | Kinetics-600 (%) | HMDB-51 (%) | UCF-101 (%) |
|---|---|---|---|
| CLAVER-B/16 (KMT) | $73.8 \pm 0.6$ | $\mathbf{54.1} \pm 2.4$ | $\mathbf{78.6} \pm 1.7$ |
| CLAVER-B/16 (KMCT) | $\mathbf{74.1} \pm 0.9$ | $54.0 \pm 2.0$ | $78.4 \pm 2.1$ |

Table 13: Comparison of KMT and KMCT on few-shot settings.

| CLAVER-B/16 | KMT | | | | KMCT | | | |
|---|---|---|---|---|---|---|---|---|
| Few-shot | K=2 | K=4 | K=8 | K=16 | K=2 | K=4 | K=8 | K=16 |
| HMDB-51 (%) | **58.6** | 63.9 | 68.0 | 72.5 | 58.3 | **64.5** | **68.6** | **72.9** |
| UCF-101 (%) | 89.7 | **92.9** | 96.1 | 98.0 | **90.0** | **92.9** | **96.6** | **98.1** |

## D EXPERIMENTAL SETTING DETAILS

**Architectures.** CLAVER-B/32 adopts ViT-B/32 ($L_V$=12, $N_h$=12, $d$=768, $p$=32) and is equipped with a KMT/KMCT transformer ($L_K$=2, $N_h$=12, $d$=768, $p$=32). CLAVER-B/16 employs ViT-B/16 ($L_V$=12, $N_h$=12, $d$=768, $p$=16), along with the KMT/KMCT transformer ($L_K$=2, $N_h$=12, $d$=768, $p$=32). CLAVER-L/14 is equipped with ViT-L/14 ($L_V$=24, $N_h$=16, $d$=1024, $p$=14) and the KMT/KMCT transformer ($L_K$=4, $N_h$=16, $d$=1024, $p$=14). Here $L_V$ denotes the layers of ViT, $L_K$ denotes the layers of Kronecker mask temporal transformer, $N_h$ refers to the number of attention heads, $d$ represents the embedding dimension and $p$ is the patch size.

**Hyperparameters.** The experiments are conducted on 8 NVIDIA 80G A100 GPUs. We present the training hyperparameters in Tab. 14. Additionally, the learning rate for updating the KMT/KMCT transformer (randomly initialized) parameters is set $10\times$ higher than the learning rate for parameters of the text encoder or image encoder. Because the text/image encoder already possesses a ability to extract high-quality text/image representations, conversely, KMT/KMCT transformer is trained from scratch. In the experiment, we freeze the parameters of the CLIP's image encoder to reduce certain computational costs, as it already has strong image feature extraction capabilities. And then we also conducted experiments on this aspect in subsequent ablation study.

Table 14: The training hyperparameters settings of experiments.

| Config | Fully-sup | Few-shot | Zero-shot |
|---|---|---|---|
| Optimizer | | AdamW | |
| Base learning rate | 12e-6 | 2e-6 | 12e-6 |
| Minimal learning rate | 12e-8 | 2e-8 | 12e-8 |
| Weight decay | | 0.001 | |
| Optimizer betas | | $\beta_1, \beta_2$ =0.9, 0.98 | |
| Batch size | | 128 (ViT-B) 32 (ViT-L) | |
| Learning rate schedule | | Cosine decay | |
| Warmup epochs | | 5 | |
| Training epochs | 60 (ViT-B) 40 (ViT-L) | 80 (20 on K400) | 0 (20 on K400) |
| Augmentation | | RandomFlip, MultiScaleCrop, ColorJitter GrayScale, Label smoothing, Mixup, Cutmix | |

**Fully-supervised experiments setting.** We conduct the fully-supervised experiments on Kinetics-400&600. During training, a sparse sampling strategy is used. The number of frames is set to 8 or 16. We spatially scale the shorter side of each frame to 256 and take a 224 crop center crop. Following, we adopt the multi-view inference with 3 spatial crops and 4 temporal clips.

**Few-shot experiments setting.** We randomly sample 2, 4, 8 and 16 videos from each class on UCF-101 and HMDB-51 constructing the training set. For evaluation, we use the first split of the test set on UCF-101 and HMDB-51.

**Zero-shot experiments setting.** We train CLAVER-B/16 with 32 frames on Kinetics-400. The same as, we apply the following two evaluation protocols in zero-shot experiments. 1) Evaluation for HMDB-51 and UCF-101. Following, the prediction is conducted on the three splits of the test data, and we report the average top-1 accuracy and standard deviation. 2) Evaluation for Kinetics-600. Following, the 220 new categories outside Kinetics-400 in Kinetics-600 are used for evaluation. The evaluation is conducted three times. For each iteration, we randomly sampled 160 categories for evaluation from the 220 categories in Kinetics-600.

## E INTERPRETIVE PROMPT DETAILS

First, we input command prompts into the ChatGPT OpenAI (2024) chatbot API to obtain a demonstration case of a small amount (in our experiment, we set it to 3 here) of detailed descriptions of actions.

**Question to ChatGPT:**

```
1  Below, I will provide you with some action nouns. Please provide a simple
2  and detailed description (explanation) about the action decomposition
3  of these action nouns. Please note that the sentence length of the
4  description should not exceed 76 words.
```

**Answered by ChatGPT:**

```
1  Certainly! Please go ahead and provide the action nouns you'd like
2  descriptions for, and I'll provide concise explanations within the
3  specified sentence length.
```

**Question to ChatGPT:**

```
1  abseiling.
```

**Answered by ChatGPT:**

```
1  Abseiling combines several actions to descend a vertical surface with a
2  rope. Climbers secure themselves with a harness and utilize a descender
3  device for controlled descent. Simple actions, like maintaining a
4  straight body position and regulating rope tension, form the basis.
5  Abseiling demands proper training, safety measures, and is popular in
6  adventure sports and rescue operations, allowing individuals to
7  experience controlled descent in various settings.
```

**Question to ChatGPT:**

```
1  air drumming.
```

**Answered by ChatGPT:**

```
1  Air drumming is a rhythmic expression where individuals simulate playing
2  drums without physical instruments. Simple actions, like mimicking
3  drumming motions in the air, combine to create this imaginative and
4  playful activity. Enthusiasts use their hands and feet to imitate
5  drumming patterns, syncing with music. It's a spontaneous, enjoyable
6  gesture often done during music listening or live performances,
7  showcasing one's connection to the rhythm without the need for actual
8  drums or drumsticks.
```

**Question to ChatGPT:**

```
1  answering questions.
```

**Answered by ChatGPT:**

```
1  Answering questions involves providing responses to queries posed by
2  others. Simple actions like active listening, comprehension, and concise
3  articulation combine in this communicative process. It is fundamental
4  in various contexts, facilitating information exchange and problem-
5  solving. Respondents draw on their knowledge and expertise to address
6  inquiries, contributing to effective communication and fostering
7  understanding between individuals or groups.
```

**Question to ChatGPT:**

```
1  Below, I will provide you with some action nouns or phrases. Please
2  provide many words and phrases that share the same central concept
3  as these action nouns or phrases but have more diverse expressions.
4  Please note that your reply should not exceed 76 words in length.
```

**Answered by ChatGPT:**

```
1  1. Running: Jogging, sprinting, dashing, hurrying, racing, pounding the
      pavement, speeding, trotting.
2  2. Eating: Dining, feasting, devouring, munching, nibbling, gobbling up,
      indulging, savoring.
3  3. Writing: Scribbling, jotting down, composing, drafting, penning,
      inscribing, scrawling, scripting.
4  4. Laughing: Chuckling, giggling, guffawing, snickering, roaring with
      laughter, belly laughing, cackling, chortling.
5  5. Reading: perusing, browsing, scanning, skimming, devouring books,
      absorbing, immersing oneself, delving into texts.
```

**Question to ChatGPT:**

```
1  Cutting in the kitchen.
```

**Answered by ChatGPT:**

```
1  Slicing, dicing, chopping, mincing, cleaving, carving, trimming,
2  preparing ingredients.
```

**Question to ChatGPT:**

```
1  driving car.
```

**Answered by ChatGPT:**

```
1  Operating a vehicle, maneuvering behind the wheel, navigating the road,
2  piloting an automobile, steering, cruising, commuting by car, motoring.
```

**Question to ChatGPT:**

```
1  Walking With Dog.
```

**Answered by ChatGPT:**

```
1  Strolling with a canine companion, ambling with a pet, promenading with a
2  pup, hiking with a furry friend, sauntering alongside a dog, wandering
3  with a four-legged buddy, leash-walking, trotting with a pooch.
```

**Question to ChatGPT:**

```
1  Below, I will provide you with some action nouns. Please describe
2  these actions based on their nouns and possible body parts involved.
3  Please note that the sentence length of your response should not exceed
4  76 words.
```

**Answered by ChatGPT:**

```
1  1. Running: The rhythmic motion of legs propelling the body forward, feet
      striking the ground with each stride, muscles contracting and
      relaxing to generate momentum.
2  2. Swimming: Arms slicing through water, legs kicking to maintain
      propulsion, torso rotating to streamline movement, lungs rhythmically
      inhaling and exhaling for oxygen.
3  3. Writing: Fingers gripping a pen or pencil, hand moving across paper or
      keyboard, muscles in the forearm and wrist coordinating precise
      movements to form letters and words.
4  4. Jumping: Legs bending at the knees, feet pushing off the ground, arms
      swinging for momentum, body soaring through the air momentarily
      before landing.
```

```
5   5. Dancing: Feet tapping, legs extending and bending, hips swaying, arms
        gesturing and flowing in sync with music, body moving rhythmically
        with expression and grace.
```

**Question to ChatGPT:**

```
1   Cutting in the kitchen.
```

**Answered by ChatGPT:**

```
1   Using a sharp knife, fingers gripping the handle, hand guiding the blade
2   through ingredients on a cutting board, wrist controlling the motion,
3   fingers curling slightly to hold the food steady, precision applied
4   to achieve desired shapes or sizes, ensuring safety and efficiency
5   during food preparation.
```

**Question to ChatGPT:**

```
1   driving car.
```

**Answered by ChatGPT:**

```
1   Gripping the steering wheel, hands adjusting position, fingers pressing
2   pedals for acceleration and braking, eyes scanning surroundings for
3   obstacles, feet coordinating between clutch, brake, and accelerator,
4   body positioned comfortably in the driver's seat, mind focused on
        navigation
5   and traffic signals, reacting swiftly to changing road conditions.
```

**Question to ChatGPT:**

```
1   Walking With Dog.
```

**Answered by ChatGPT:**

```
1   Leash in hand, fingers securing grip, arm relaxed as it swings alongside
2   the body, legs moving in tandem with the dog's pace, feet stepping
3   forward with purpose, eyes attentive to the dog's behavior and
        surroundings,
4   occasional stops for sniffing or marking, a bond of companionship
5   evident in synchronized movement.
```

Subsequently, as illustrated in Fig 4, we input the examples generated by ChatGPT OpenAI (2024) into LLaMA-3 Abhimanyu Dubey et al. (2024) multiple times in the format of ***Command + Examples + Action Concept***, automatically generating diverse text descriptions about the ***Action Concept*** through the program. When dealing with datasets containing a large number of categories, this scheme can greatly save labor costs. In addition, it should be noted that LLaMA-3 offers controllable parameters to control the randomness ($\tau$) and diversity ($p$) of output texts. We set $\tau$ and $p$ to 0.90 and 0.95, respectively to ensure that there are significant variations in output content for the same input, while maintaining consistency in the central concept.

## F ANALYSIS OF SYNTHETIC VIDEO TESTING

We use the original prompts of synthetic videos (several action-related examples generated by Imagen Saharia et al. (2022) and Sora Brooks et al. (2024)) as their corresponding text descriptions, and show Top-5 most relevant texts in Fig. 9. The results shows the robustness and generalization of CLAVER.

For cases generated by Imagen, in Fig. 9 **Upper**, we denote the Kinetics-400 action text set as $\mathcal{A}$. The action category "lift a cup" in Fig. 9 (a) does not appear in $\mathcal{A}$, while, (b) "washing dishes", (c)

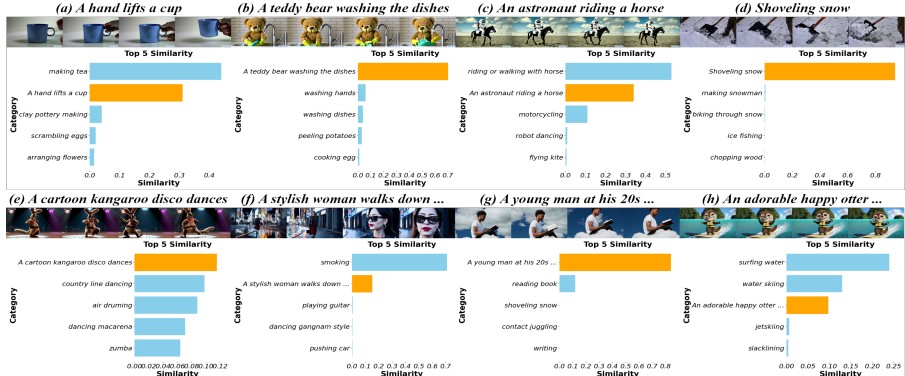

Figure 9: Testing on synthetic videos from Imagen (**Upper**) and Sora (**Lower**). The bar chart represents the Top-5 similarity texts, orange bar denotes the original prompts of synthetic videos.

"riding horse", (d) "shoveling snow" do appear in $\mathcal{A}$. All videos do not come from the real-world, which are synthetic. We observe that all the original prompts appear in the Top-5 similarity (more precisely, Top-2). Specifically, for (a), "making tea" has the highest similarity, possibly due to the presence of cups and human hands in the scene. In addition, the original text has the second-highest similarity. In (b), apart from the original prompt in Top-1, other texts such as "washing dishes" convey the same meaning as the original prompts but lack a description of the subject ("teddy") of the action. "Washing hands" and "peeling potatoes" may involve actions and scenes like hand movements and sinks, thus exhibiting high similarity. For (c), although the one with the highest similarity is not "An astronaut riding a horse" , "riding or walking horses" is the exactly describes the action in the video, with the only difference being the lack of the subject of the action. For (d), "Shoveling snow" belongs to $\mathcal{A}$, and since the original prompt of the video is "Shoveling snow", its similarity is very high, exceeding 90%.

For the cases generated by Sora, as depicted in Fig. 9 **Lower**, (e) "disco dances" concept is not in $\mathcal{A}$ but has the highest similarity. Besides, other action concepts in the Top-5 ("country line dancing","air druming", "dancing macarena","zumba") are all dance-related. (f) is a relatively challenging sample here. Although the ground truth description is located in the second, the "smoking" in Top-1 is entirely unrelated to the content of the video, because the original prompt of the video primarily describes the scene and the action-related text is relatively short. In addition, the shot is a process from far to near, which may lead the model to arrive at an unreasonable Top-1. In (g), the Top-2 descriptions are original prompt and "reading book", both strongly related to the action in video. For (h), the original prompt only rank third, however, Top-1 "surface water" and the second "water skimming" are descriptions strongly related to the action in video.

## G  DATASET DETAILS

***Kinetics-400&600.*** The Kinetics dataset consists of 10-second video clips collected from YouTube. Specifically, Kinetics-400 Kay et al. (2017) consists of approximately 240k training videos and 20k validation videos with 400 categories, while Kinetics-600 Carreira et al. (2018) is an extension of Kinetics-400, consisting of approximately 410k training videos and 29k validation videos with 600 categories.

***UCF-101.*** UCF-101 Soomro et al. (2012) consists of 101 action categories, over 13k clips and 27 hours of video data. The database comprises realistic user uploaded videos containing camera motion and cluttered backgrounds. The training and test data are divided into three splits.

***HMDB-51.*** HMDB-51 Kuehne et al. (2011) is a collection of realistic videos from various sources, including movies and web videos. It is composed of 6,766 video clips from 51 action categories, with each category containing at least 101 clips. The dataset is divided into three splits for training and test data. In each split, each action class has 70 clips for training and 30 clips for testing.

***Synthetic Videos.*** We select some action-related videos from the demo cases generated by Imagen Saharia et al. (2022) and Sora Brooks et al. (2024), and employ the original prompts and their corresponding synthetic videos as a pair of test samples.

## H  MORE ATTENTION HEAT MAP VISUALIZATION

We visualize several spatiotemporal attention map of samples from Kinetics400, Imagen, Sora.

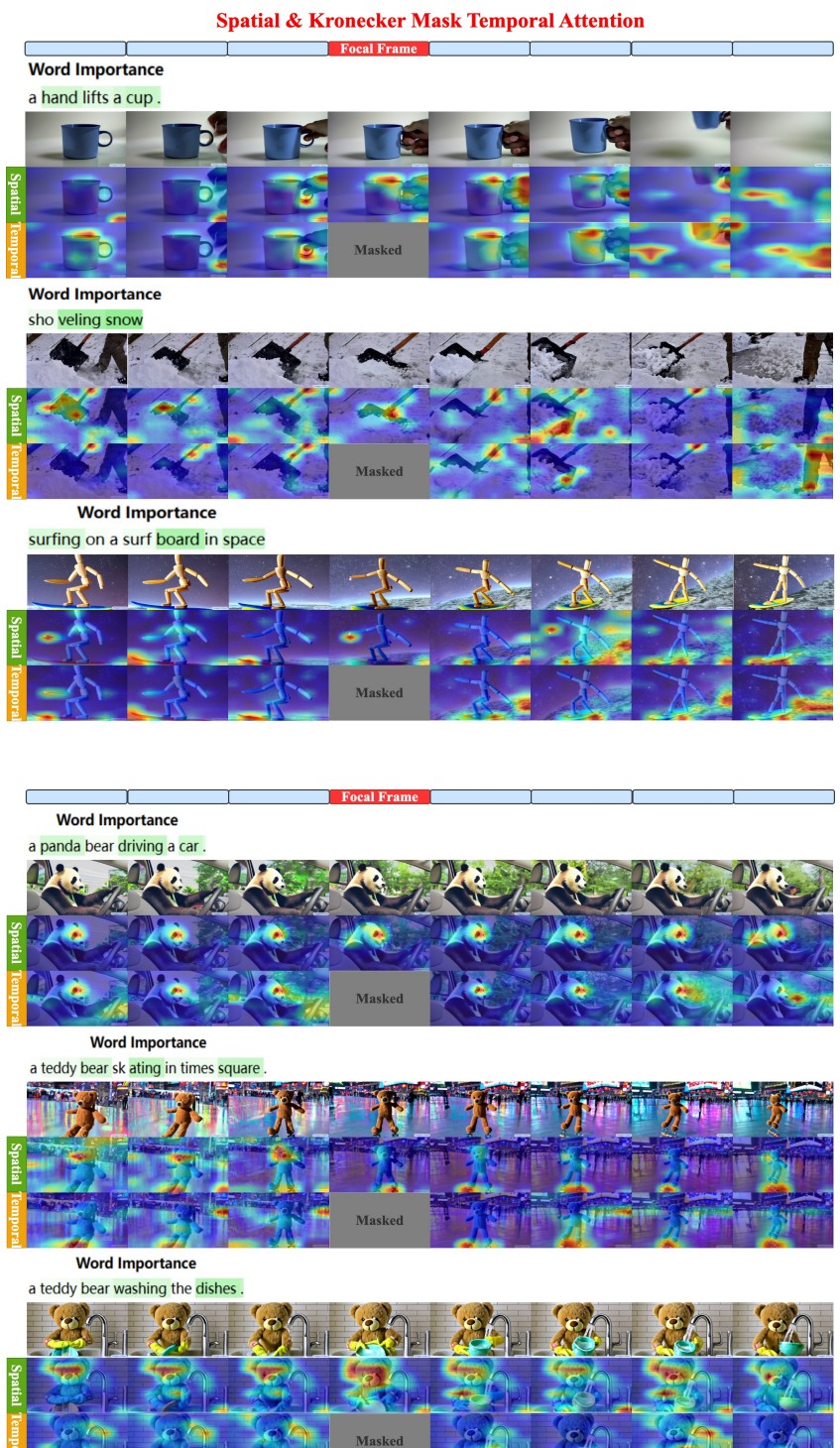

Figure 10: Attention map of synthetic videos from Imagen.

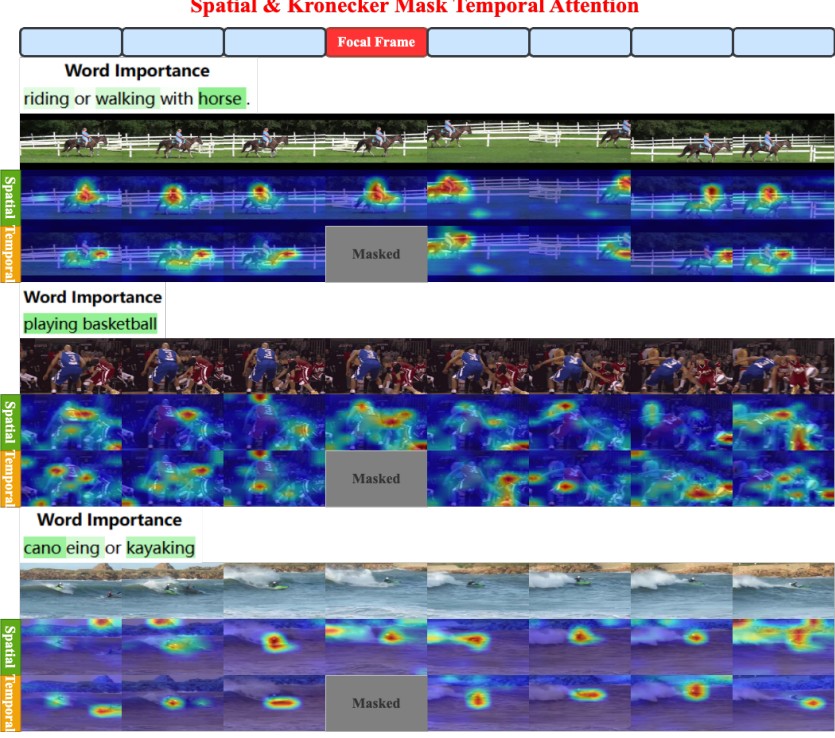

Figure 11: Attention map of videos from Kinetics400.

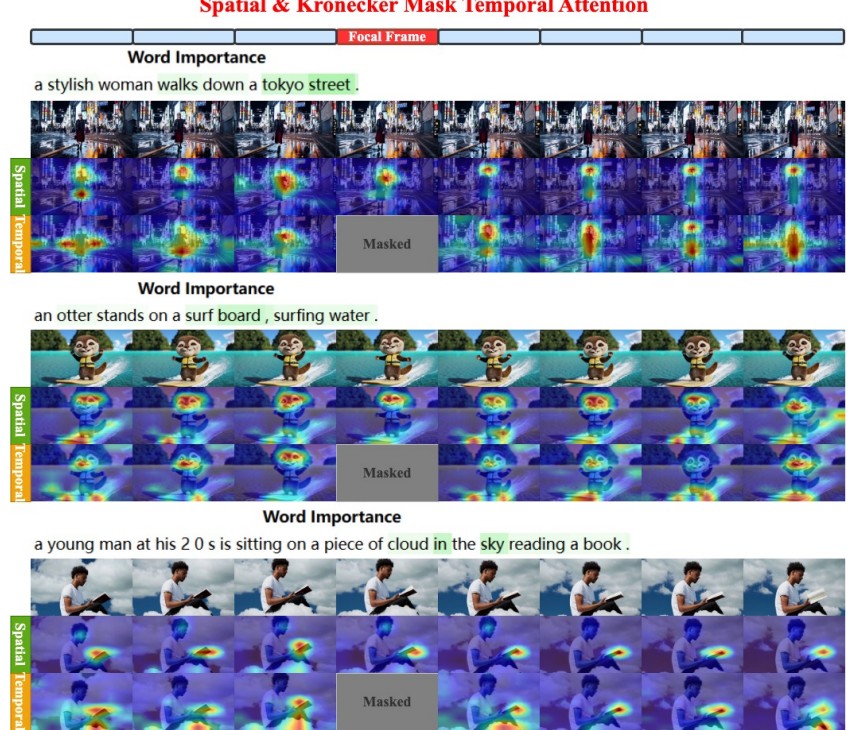

Figure 12: Attention map of synthetic videos from Sora.

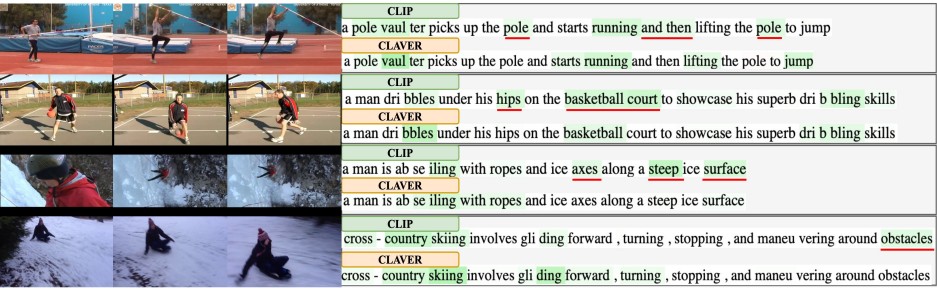

Figure 13: More visualization examples of transition of CLIP's attention on nouns to CLAVER's preference on verbs.

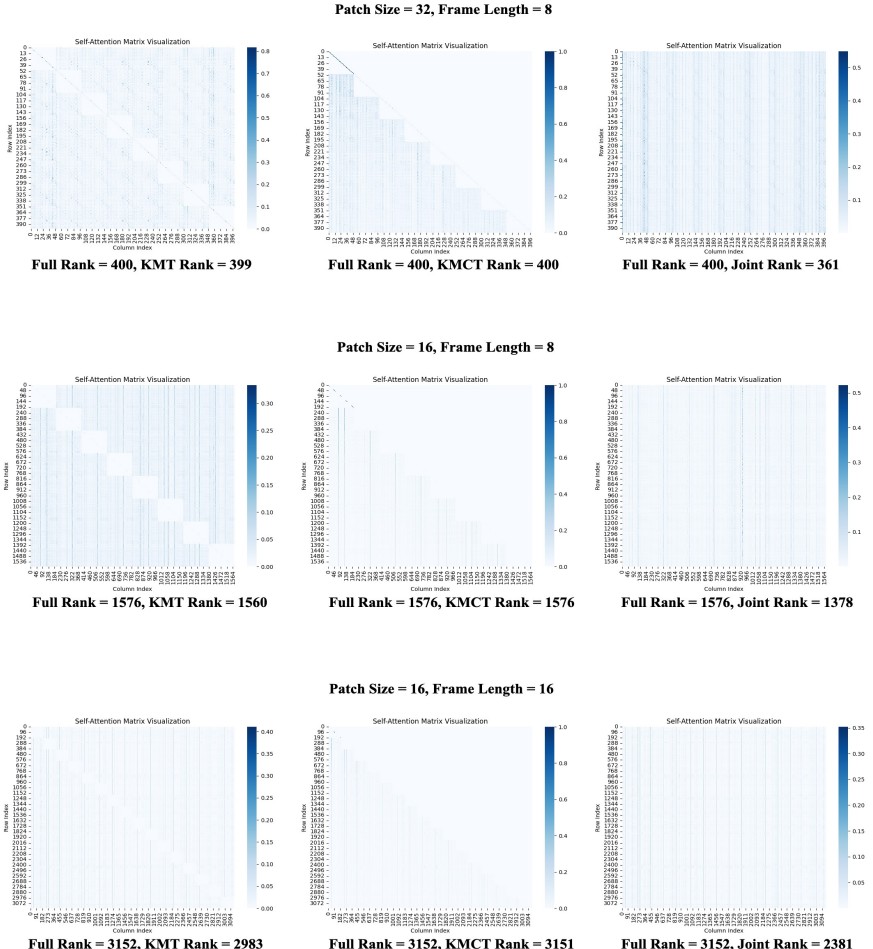

Figure 14: The rank of Kronecker temporal attention (KMT), Kronecker causal temporal attention (KMCT) and joint attention (Joint) on different patch size and frame length. (The calculation of rank is done via torch.linalg.svd)

