# OpenReview forum: "Kronecker Mask and Interpretive Prompts are Language-Action Video Learners"
_ICLR.cc/2025/Conference — ICLR 2025 Poster_

### Official Review · Reviewer_sCy3 · 2024-11-01

**Soundness:** 3
**Presentation:** 3
**Contribution:** 3
**Rating:** 8
**Confidence:** 3

**Summary:**

This paper proposes a Contrastive Language-Action Video Learner (CLAVER) to efficiently adapt both the visual and textual branches of the CLIP for improving video action recognition. CLAVER introduces the Kronecker mask for temporal modeling to capture the long-range and wide-range dependencies among frames. Moreover, CLAVER resorts to LLMs to generate interpretive prompts of actions to facilitate the alignment of action behaviors and verbs. Experimental results on multiple benchmarks, such as Kinetics-400 and Kinetics-600, show the superiority of the proposed method.

**Strengths:**

1 The proposed method incorporating the Kronecker mask and interpretive action prompts is demonstrated to be effective in supervised, few-shot, and zero-shot settings.

2 The visualization results sufficiently show that the proposed method captures better spatial-temporal attention as well as focuses more on verbs.

3 The proof of KMCTA can guarantee full rank is provided in detail in the Appendix.

4 The paper is well-written with figures and tables nicely presented.

**Weaknesses:**

Overall, I think this paper introduces an effective and generalizable mechanism for improving video action recognition. The proposed method is novel and supported by extensive experimental results in both the main text and Supp. I only have some small concerns as follows:

1 Since the LLM tends to introduce some task-agnostic information in its output, will this noise impact the actual action recognition performance? I would like to see more analysis and examples here. Moreover, the proposed interpretive prompts look like a plug-and-play module. When it is combined with other existing models, will it contribute to performance improvement consistently?

2 Can the authors give a complete analysis with examples to show when the proposed method may fail? This will help better understanding and deeper analysis of the proposed method.

3 The font size in Figure 6 is too small for reading. The authors should modify it to give a better presentation. Moreover, Table 14 in Supp has the wrong format.

**Questions:**

See details in the weaknesses.

---

> ### Author Response · Authors · 2024-11-17
> **Author's Response (1/2)**
>
> Thanks for your encouraging comments and constructive feedback.
>
> **1) Concern about the task-agnostic information in LLM’s output:**
>
> Thank you for raising this thoughtful question. Indeed, LLM outputs can sometimes include task-agnostic content (referred to as noise), which we have encountered in our experiments. To mitigate this issue, we implemented certain measures, such as the format prompt mentioned in Sec. 3.3 and illustrated in Fig. 4. This format prompt guides LLaMA in generating interpretive prompts by structuring the format of **Commands**+**Examples**+**Action Concept**, providing a certain degree of control over the generated text. The **Examples** field, crafted through human-LLM interaction, aligns the text more closely with our expectations. The **Commands** field instructs LLaMA to adhere to the style of the given **Examples**. However, while this operation reduces bias, it does not entirely eliminate noise. To further filter low-quality or unexpected outputs, we employed simple approaches like human selection and asking LLMs to self-evaluate whether the generated text aligns with the expected **Examples** style, retaining only outputs with positive (Yes) confirmations.
>
> Additionally, we observed that while some interpretive prompts improve performance, others may degrade it. Noise may have negative effects, often influenced by the design of interpretive prompts themselves. Taking the action ‘abseiling’ as an example, three perspectives of interpretive prompts used in Fig. 4 include:
>
> >**Description of action decomposition** *Abseiling combines several actions to descend a vertical surface with a rope. Climbers secure themselves with a harness and utilize a descender device for controlled descent. Simple actions, like maintaining a straight body position and regulating rope tension, form the basis*
> >
> >**Synonyms conversion** *Descending, rappelling, downclimbing, descending a rock face, climbing down*
> >
> >**Bringing into body parts** *Abseiling involves the following actions. Feet planted securely on rock face, hands gripping rope for balance and support, arms stretched overhead, torso leaning backward, body fluidly transitioning between maintaining balance and swinging downward, legs straight or bent for control*
>
> Furthermore, we also explored some other designs, such as describing a scene corresponding to an action concept:
>
> >**Describe a scene involved an action** *A scene involves abseiling. A climber slowly makes their way down the sheer rock face, their footing steady and deliberate. Hands grip the rope for support as they rappel, **the wind whipping past their face, a myriad of emotions flitting across their face. With each drop, they gain control, their confidence in the safety line growing with every passing moment***
>
> However, the text in bold italics is not related to the action content but rather to the emotions of the subject performing the action. Introducing such interpretive prompts can negatively affect model performance.
>
> Our motivation for designing three interpretive prompts is that **action decomposition**: helping distinguish complex and basic actions, and similar actions, as not all action concepts are of equal status, **synonym conversion**: improving zero-shot generalization, and **introducing body parts**: helping localize the region where action occurs. Their effectiveness was verified through extensive experiments. However, **describing a scene involved an action** may be easier to introduce potential noise because the concept corresponding to an action scene may be broader and more unconstraint.
>
> Interpretive prompt (IP) is a promising general module for action recognition and understanding. We integrate it with previous state-of-the-art methods and observe the improvements, as shown in the following table:
> | Methods | Top-1(%) | Top-5(%) |
> |------------|------|------|
> | X-CLIP-B/32$_{8f}$  | 80.1 | 94.8 |
> | X-CLIP-B/32$_{8f}$ + IP | **80.9** | **95.2** |
> | ILA-B/32$_{8f}$  | 80.6 | 94.9 |
> | ILA-B/32$_{8f}$ + IP | **81.1** | **95.4**  |

---

> > ### Author Response · Authors · 2024-11-17
> > **Author's Response (2/2)**
> >
> > **2) About the failure case analysis:**
> >
> > Thank you for suggesting a deeper analysis of failure cases. Certain scenarios, such as videos involving transitions between near and far shots or when core of textual descriptions deviate significantly from the action concept, pose challenges for our method.
> >
> > For example, as shown in Appendix H, Fig. 11 (f), presents a relatively challenging case where the ground truth description is located in the Top-2 with a small value, but the Top-1 prediction, 'smoking', is entirely unrelated to the video content. The ground truth text primarily describes the scene: “A stylish woman walks down a Tokyo street filled with warm glowing neon and animated city signage. She wears a black leather jacket, a long red dress, and black boots and carries a black purse...” Here, the action-related content is minimal. Moreover, the transition from far to near shots likely contributes to the model’s misunderstanding. We believe this issue stems from limited training data with near-far shot transitions, and the model’s input being sparse sampling (8 to 16 frames) from the original videos.
> >
> > **3) Regarding issues of Figure and Table:**
> >
> > We have corrected these issues and updated them in the rebuttal revision. Please check the corresponding location in the updated PDF.

---

> > > ### Comment · Reviewer_sCy3 · 2024-11-25
> > > **Response to authors**
> > >
> > > The authors have addressed my concerns by providing additional experimental results and the modified manuscript. I will keep my rating on this paper.

---

> > > > ### Author Response · Authors · 2024-11-27
> > > >
> > > > Thank you for your positive feedback and continued recognition of our paper! We are appreciative of your thoughtful review and delighted that we have addressed your concerns.

---

### Official Review · Reviewer_2oCA · 2024-11-03

**Soundness:** 3
**Presentation:** 3
**Contribution:** 2
**Rating:** 6
**Confidence:** 4

**Summary:**

The paper introduces a novel framework called Contrastive Language-Action Video Learner (CLAVER), aimed at enhancing the adaptation of CLIP for video understanding, particularly in action recognition tasks. This research addresses a critical gap in existing methodologies, which often focus on modifying either the textual or visual components of CLIP in isolation. The authors argue that a dual adaptation of both branches is essential for effective video analysis. The key contributions are twofold: (1) Kronecker Mask Attention that significantly improves temporal modeling in video data (2) The authors utilize large language models to generate rich, sentence-level prompts that focus on action verbs, steering the model's learning towards a deeper understanding of dynamic behaviors rather than static objects. The extensive experiments across multiple benchmarks, showcasing the effectiveness and robustness of CLAVER compared to existing methods.

**Strengths:**

* The paper is well-written and easy to understand. The figure illustration and captions are informative.
* The proposed Kronecker mask temporal attention and Kronecker mask causal temporal attention schemes seem interesting. The authors have proved its effectiveness for temporal modeling
* Experimental results on four widely used benchmarks show that the proposed model could achieve superior performance with the existing contrastive baselines.

**Weaknesses:**

* The performance gain is marginal. The advantages on multiple benchmarks compared to the current state-of-the-art methods do not show a significant improvement, which somewhat undermines the effectiveness of this approach. It remains unclear whether the enhancements stem from the hyperparameter settings or other tricks.
* Can the authors provide details on how much additional computational cost and efficiency are incurred by adding the extra Kronecker mask temporal attention compared to the original CLIP method?

**Questions:**

Refer to weakness.

---

> ### Author Response · Authors · 2024-11-17
> **Author's Response**
>
> Thanks for your feedback and for highlighting the strengths of our work. Below, we address your concerns in detail.
>
> **1) Regarding the performance:**
>
> We appreciate your concern regarding the potential influence of hyper-parameter settings. In Appendix D, Tab. 14, we provide a detailed hyper-parameter setting. Compared to the previous state-of-the-art methods, we slightly adjust the learning rate and epochs to observe a suitable convergence trend of the model, without any other modifications, also no additional tricks are employed. Moreover, CLAVER enhances not only the classification accuracy but also the interpretability of the model’s predictions, as evidenced by the spatial-temporal attention visualizations in Fig. 7 and Appendix Fig. 12-14. Beyond fully-supervised performance metrics, CLAVER demonstrates superior generalization, particularly in zero-shot settings where domain shifts and unseen classes pose significant challenges. We would like to stress that small yet consistent improvements on highly competitive benchmarks such as Kinetics-400, Kinetics-600, UCF-101, and HMDB-51 are non-trivial, given the saturation of these datasets.
>
> To further validate the effectiveness of each module in our method, we have conducted ablation studies (see Tab. 6) that isolate the contributions of KMTA and interpretive prompts. These studies confirm that the improvements stem primarily from our proposed components rather than external factors. In addition, as shown in the table below, compared with the previous state-of-the-art ILA, our method has smaller parameter size and FLOPs, while the performance is better, which also indicates the efficiency of our method.
>
> **2) About the computational cost and efficiency:**
>
> We provide a detailed comparison of FLOPs, parameter size, and inference time for CLIP, ILA, XCLIP and our CLAVER in the table below. In addition, our model is trained on 4 NVIDIA A100 GPUs (CLAVER-B-32, CLAVER-B-16) or 8 NVIDIA A100 GPUs (CLAVER-L-14) for 3-5 days (on Kinetics-400 dataset). We test the inference time using a NVIDIA GeForce RTX 4090 GPU.
>
> The parameter size of CLAVER (B/32, B/16, L/14) are 1.3x that of CLIP (B/32, B/16, L/14), the values of FLOPs increase by 1.2x, and the increment of inference time are (1x, 2x, 3.6x) respectively. The parameter size of CLAVER (B/32, B/16, L/14) are very similar to X-CLIP (B/32, B/16, L/14), with FLOPs being 1.3x and inference time increments of (0.75x, 1.3x, 1.6x), respectively. In terms of performance, CLAVER is much better than X-CLIP. The parameter size of CLAVER (B/32, B/16, L/14) are 0.77x\~0.8x that of ILA (B/32, B/16, L/14), with FLOPs being 0.8x\~0.86x and inference time increments of (0.2x, 0.4x, 0.7x), respectively. CLAVER is better than ILA in both terms of performance and computational cost.
>
> | Model                    | FLOPs(G) | Params(M) | Inference time(ms) |
> |----------------------|----------|-----------|--------|
> | CLIP-B/32$_{8f}$   | 25.51    | 84.23     | 7.47  |
> | CLIP-B/16$_{8f}$   | 92.10    | 82.46     | 8.81  |
> | CLIP-L/14$_{8f}$   | 419.52  | 258.72   | 20.13 |
> | X-CLIP-B/32$_{8f}$ | 25.64   | 99.72    | 9.54   |
> | X-CLIP-B/16$_{8f}$ | 92.39   | 97.95    | 12.46 |
> | X-CLIP-L/14$_{8f}$ | 420.38  | 302.87  | 46.39 |
> | ILA-B/32$_{8f}$    | 40.24    | 133.08   | 33.65   |
> | ILA-B/16$_{8f}$    | 150.59  | 131.30   | 36.89   |
> | ILA-L/14$_{8f}$    | 647.70   | 395.65    | 104.03 |
> | CLAVER-B/32$_{8f}$ | 33.06   | 103.11  | 7.18   |
> | CLAVER-B/16$_{8f}$ | 121.85  | 101.35  | 16.42 |
> | CLAVER-L/14$_{8f}$ | 557.50  | 325.87 | 72.81   |
>
> \* Note that the FLOPs here are calculated with a batch size of 1, 8 frames and a maximum sentence length of 77. The tools used to calculate these metrics are thop.profile and torch.utils.benchmark, where the inference time is the average of 100 runs.

---

> > ### Author Response · Authors · 2024-11-18
> > **Reminder**
> >
> > Fig. 15, which we add in the rebuttal revision, is at the end of the Appendix.

---

> > ### Comment · Reviewer_2oCA · 2024-11-25
> > **Reply to authors**
> >
> > Thanks to the authors for addressing my concerns and questions. I will keep the positive rating on this paper.

---

> > > ### Author Response · Authors · 2024-11-27
> > >
> > > Thank you for your positive feedback of our paper! We are appreciative of your thoughtful review and delighted that we have addressed your concerns.

---

### Official Review · Reviewer_RydQ · 2024-11-07

**Soundness:** 4
**Presentation:** 4
**Contribution:** 3
**Rating:** 6
**Confidence:** 4

**Summary:**

The author proposes CLAVER: a Contrastive Language-Action Video Learner, to shift CLIP’s focus for video action recognition tasks from the alignment of static visual objects and nouns to dynamic action behaviors and abstract verbs. They introduce CLAVER with generalizable Kronecker Mask Temporal Attention (KMTA) to expand the temporal receptive field for each token and serve as an effective spatiotemporal heterogeneity inductive bias, mitigating spatiotemporal homogenization. In addition, they utilize large language models to generate interpretive prompts of actions, which shift the model’s focus towards verb comprehension. They conduct extensive experiments on well-known action recognition datasets like Kinetics-400, Kinetics-600, HMDB-51, and UCF-101 demonstrating its competitive performance across models in full-shot, few-shot, and zero-shot settings.

**Strengths:**

•	Innovative Kronecker Mask Attention: The Kronecker mask temporal attention and Kronecker mask causal temporal are innovative and effective for temporal modeling. Compared with previous commonly used approaches such as joint attention, spatial attention, pipeline temporal attention, etc., it allows each patch at timestamp t to interact with all other patches and expands the temporal receptive field width of each token. It also alleviates the impact of spatiotemporal homogenization.

•	Leveraging large language models to create diverse, sentence-level prompts for action verbs is an effective approach. The author focuses on action decomposition, synonym conversion, which is an inspiring idea of “facilitating the alignment of action behaviors and verbs” by decomposing complex actions into basic and conveying the same core concept with varied expressions.

•	This paper is well-written and provides a clear overview of CLAVER, with Kronecker Mask Temporal Attention and interpretive prompts. It conducts extensive experiments across several commonly used benchmarks with previous state of the art models as comparison and shows competitive performance, especially in few-shot and zero-shot settings, and provides comprehensive ablation studies to clearly claim the strength of CLAVER by parts.

**Weaknesses:**

•	Insufficient performance difference between KMTA and KMCTA. KMCTA aims to alleviate the low-rank bottleneck and the author claims that it is important to avoid the low-rank bottleneck to improve the representation power of transformer architecture by giving a formal proof. In the latter ablation studies, the author also successfully shows that KMCTA is profoundly affected by both PreTE and PostTE shuffling, which suggests that KMCTA possesses varying degrees of ability in mitigating spatiotemporal homogenization. However, based on the experiment results in Table. 1 and Table. 7, the difference between KMTA and KMCTA is marginal and we also lack experiments on zero-shot, few-shot settings for CLAVER with KMCTA, which I didn’t see how KMCTA solved the bottleneck according to the current experiments setting.

•	The pipeline of interpretive prompts. The author introduces an interesting pipeline of interpretive prompts by performing action decomposition, synonym conversion, and involving body parts. Then, they provide the prompt to LLaMA-3 for text completion. However, based on the illustration in Figure 4, and the textual description, I feel confused about how to make up these three steps from the left side of the figure to the right side of the figure into format prompts to LLaMA in order to perform text completion.

•	Qualitative analysis for word importance needs further proof. Based on the visualization for CLIP, X-CLIP in Figure. 5, while the author make a claim that CLIP tends to focus on nouns, whereas CLAVER prefers verbs, it’s hard for me to recognize the difference between CLIP and CLAVER’s visualization. As a result, the effectiveness of interpretive prompts for noun concept to verb concept transition might need further explanation.

**Questions:**

In addition to the questions raised in the weakness section, there are a few minor questions.

•	In Figure 4, the transaction from the left side to the right side might need further explanation. In Figure 5, the example might not fully explain the claim for word importance, please consider making another effective example.

•	In Table 1, we can see the most powerful model for CLAVER is the CLAVER-L/14(KMT/KMCT). However, when conducting later experiments on Kinetic-600, and other few-shot, zero-shot experiments, the author uses CLAVER-B/16 (KMT) but not the strongest model in Table.1 even if it can follow the same setting. I’m wondering the reason why the author didn’t use their best model for all experiments.

---

> ### Author Response · Authors · 2024-11-17
> **Author's Response**
>
> Thanks for your encouraging comments and thoughtful suggestions. Below, we address your concerns in detail.
>
> **1) Insufficient performance difference between KMTA and KMCTA and how KMCTA solved the bottleneck:**
>
> Thanks for your attention to this issue. Tab. 8 reflects the impact of low-rank bottleneck that may arise from different patch sizes and frame lengths, comparing joint attention (JA), KMTA, and KMCTA. Perhaps due to the font size being too small, it may not be as noticeable, and we have fixed this issue in the rebuttal revision. Tab. 8 indicates that reducing patch size increases the number of tokens but does not highlight the low-rank bottleneck for KMTA and JA, as smaller patch sizes result in finer-grained features. JA, KMTA and KMCTA all achieve better performance. In contrast, when we increase the frame length, only the performance of KMCTA can steadily improve, and the performance increment of KMCTA is the largest. You can refer to related analysis in Sec. 4.3 (line 459-468) in main text.
>
> However, this phenomenon is not so obvious in Tab. 1, likely because the Kinetics-400 is large and diverse enough, while the size of HMDB-51 and UCF101 are relatively smaller. Thus, in situations where data is limited, the low-rank bottleneck becomes more pronounced. The limited data is insufficient to support the model's full learning, which cannot effectively learn the diversity features in the data distribution, leading to the degradation of the attention matrix.
>
> Thanks also for your reminder. We evaluate the performance of zero- and few-shot experiments on Kinetics-600, HMDB-51, and UCF-101 with CLAVER-B/16 (KMCT) configuration. The comparison of KMT and KMCT as shown in the following tables:
>
> | Zero-shot  | Kinetics-600 (%) | HMDB-51 (%) | UCF-101 (%) |
> |----|----|----|----|
> | CLAVER-B/16 (KMT) | 73.8 $\pm$ 0.6  | **54.1** $\pm$ 2.4 | **78.6** $\pm$ 1.7 |
> | CLAVER-B/16 (KMCT) | **74.1** $\pm$ 0.9  | 54.0 $\pm$ 2.0 | 78.4 $\pm$ 2.1 |
>
> |  CLAVER-B/16  |   KMT  |   |   |   |   | KMCT   |   |   |   |   |
> |---------------|---|---|---|---|---|---|---|---|---|---|
> |   **Few-shot**  |  K=2 | K=4  | K=8 | K=16 |  |   K=2 | K=4  | K=8 | K=16 |
> |    HMDB-51 (%)  | **58.6**  | 63.9 | 68.0 | 72.5  |  | 58.3  | **64.5** | **68.6** | **72.9** |
> |    UCF-101  (%)  | 89.7 | **92.9** | 96.1 | 98.0  |  | **90.0**  | **92.9** | **96.6** | **98.1** |
>
> In the zero-shot scenario, the performance of KMCTA and KMTA has its own wins and losses, which is due to the lack of further fine-tuning, and the model's ability is relatively dependent on previous training. In the few-shot scenario, we can observe that in most cases, the performance of KMCTA is better, which further indicates that when the scale of training data is limited, the low-rank bottleneck is more significant. There are also some related studies that have been involved in this phenomenon:
> >On the Efficiency of Transformers: The Effect of Attention Rank
> >
> >Efficient Transformers: A Survey
>
> **2) The pipeline of interpretive prompts:**
>
> The **Commands** field and **Examples** field are prepared in advance. Assuming we have 400 action classes (taking K400 as an example), we denote the original text of each class as **Action Concept** field (The colors of **Commands**, **Examples**, and **Action Concepts** fields correspond to the texts in the box). Then, we concatenate these three fields in the order of **Commands** + **Examples** + **Action Concept** to form a format prompt. We input it into LLaMA and obtain the corresponding interpretive prompt for that action class, which is used for training CLAVER. Three different interpretive prompt perspectives correspond to different **Commands** and **Examples** field contents, and the output descriptions for each interpretive prompt perspective are stored in a file for loading during training. Of course, the entire interpretive prompt generation process and training can also be conducted online. In order to improve training efficiency, we create and store the interpretive prompts before conducting training.
>
> **3) Further analysis for word importance:**
>
> We provide some representative visualizations in the Appendix (Fig. 15) to illustrate. We use sentence-level descriptions, and it can be observed that CLIP tends to prefer nouns in sentences, while CLAVER tends to prefer verbs or phrases that combine verbs and nouns.
>
> **4) Regarding why not use the CLIP-L/14 setting for all experiments:**
>
> Because many previous CLIP-based action video recognition works, like Action-CLIP [1], X-CLIP [2], and ASU[3], etc., employ the CLIP-B/16 configuration on Kinetic-600, HMDB-51, UCF-101 few-shot, zero-shot experiments. In order to make a fair comparison with these methods, we also follow the same configuration. Thus, we use the CLAVER-B/16.
> >[1] ActionClip: A New Paradigm for Video Action Recognition
> >
> >[2] Expanding Language-Image Pretrained Models for General Video Recognition
> >
> >[3] Video Action Recognition with Attentive Semantic Units

---

> ### Author Response · Authors · 2024-11-23
> **Update Fig. 4, Fig. 6 and Fig. 15**
>
> In the rebuttal revision, we recalibrate Fig. 4 and Fig. 6. Additionally, we add a new representative word importance visualization, which is Fig. 15 placed at the end of the Appendix, to show the transition of word importance from CLIP's attention on nouns to CLAVER's preference on verbs.

---

> ### Comment · Reviewer_RydQ · 2024-11-26
>
> Thanks to the authors for taking great efforts on the rebuttal. Regarding my questions, I think:
> The pipeline of interpretive prompts becomes clear to me and new examples for qualitative analysis of word importance are great. Also, the explanation for Table 1 remains using CLAVER-B/16 is meaningful.
> However, I still feel like the claim for KMCTA aims to alleviate the low-rank bottleneck remains marginal for me and the current table cannot fully explain the performance.
> Overall, I think it's a relatively great paper with a Contrastive Language-Action Video Learner (CLAVER) to efficiently adapt both the visual and textual branches. So I will keep my positive rating.

---

> > ### Author Response · Authors · 2024-11-29
> > **Response to Reviewer**
> >
> > Thank you for your positive feedback and recognition of our work! We appreciate your thoughtful questions/comments and the detailed examination of our rebuttal. We deeply sense your sincerity, which serves as a great inspiration for us.
> >
> > Regarding your concern about the contribution of KMCTA in mitigating low-rank bottleneck compared to KMTA, we provide a visualization of the rank of the temporal attention matrix of KMTA, KMCTA and joint attention (JA) in **Fig. 16** (updated in the new rebuttal revision and located at the end of the Appendix). We calculate the rank of these matrices through SVD. We summarize the observations of **Fig. 16** into the following table (patch size is denoted as **p**, frame length is denoted as **f**):
> >
> > | Attention    | SVD Rank/Full Rank (Ratio) |
> > |----------------------|----------|
> > | **p=32, f=8**  |   |
> > | JA   | 296/400 (0.74) |
> > | KMTA   | 397/400 (0.99) |
> > | KMCTA | 400/400 (1.0)  |
> > | **p=16, f=8**  |     |
> > | JA   | 1198/1576 (0.76) |
> > | KMTA   | 1312/1576 (0.83)  |
> > | KMCTA | 1505/1576 (0.95)  |
> > | **p=16, f=16**  |     |
> > | JA   | 1568/3152 (0.50) |
> > | KMTA   | 2480/3152 (0.79) |
> > | KMCTA | 2992/3152 (0.95) |
> >
> > This indicates that in general, the rank relationship among the three attention types is **Rank(KMCTA) > Rank(KMTA) > Rank(JA)**. This demonstrates that KMCTA indeed increases the rank of attention matrix, effectively.
> >
> > We would like to provide a thorough interpretation of Tab. 8 for further discussion. The performance in Tab. 8 shows that:
> >
> > + **Longer Frame Impact on Inter-Method Performance**: KMCTA performs better than KMTA and JA at longer frame settings, HMDB-51, p=32, f=16, KMCTA (69.4/+1.4) > KMTA (69.1/+1.1) > JA (68.0/+0.0), UCF-101, p=16, f=16, KMCTA (72.8/+0.7) > KMTA (72.3/+0.2) > JA (72.1/+0.0).
> >
> > + **Longer Frame Impact on Intra-Method Performance**: When increasing the frame length from 8 to 16, the performance gain of KMCTA is the greatest, HMDB-51 KMCTA (+0.7) > KMTA (+0.3) > JA (+0.1), UCF-101 KMCTA (+0.6) > KMTA (-0.9) > JA (-0.1),
> >
> > Additionally, Tab. 8 indicates that reducing the patch size increases the number of tokens, the rank ratio decreases in Fig. 16, but none of the three attention types is affected by low-rank bottlenecks, instead their performance increases. This suggests that smaller patch sizes are advantageous in capturing finer grained features, which surpasses the low-rank bottleneck issues they may bring, thereby improving performance. In contrast, increasing the frame length primarily benefits KMCTA, and the amplitude and increment of performance are both optimal. It can be seen that for vision/video tasks, low-rank bottleneck may be caused by multiple factors (e.g., patch size, frame length), but may not necessarily have a negative impact on performance.
> >
> > Moreover, we also want to use other studies on alleviating low-rank bottlenecks in visual tasks as examples to illustrate that alleviating low-rank bottlenecks does not necessarily lead to a booming increase in performance. For instance, Flatten Transformer [1], which aims to alleviate the low-rank bottleneck problem in Vanilla and Linear Attention Vision Transformers [2,3]. According to Tab. 1, Tab. 2, and Fig. 6 in [1], the performance improvement brought about by alleviating the low-rank bottleneck ranges approximately from -0.1 to 0.8, which is quite close to the performance improvement amplitude in our experiments. Therefore, this level of improvement may be a common phenomenon in vision tasks.
> >
> > > [1] FLatten Transformer: Vision Transformer using Focused Linear Attention
> > >
> > > [2] An image is worth 16x16 words: Transformers for image recognition at scale
> > >
> > > [3] EfficientViT: Enhanced linear attention for high-resolution low-computation visual recognition
> >
> > **_The above observations and viewpoints indicate that the most intuitive reflection of low-rank bottlenecks may be the rank of the attention matrix, followed by performance_**.
> >
> > Sincerely, we appreciate your discussion and positive comments, as well as the joint efforts you made and the valuable time you spent to improve this work better.

---

### Official Review · Reviewer_rNM4 · 2024-11-10

**Soundness:** 3
**Presentation:** 3
**Contribution:** 2
**Rating:** 6
**Confidence:** 5

**Summary:**

This paper focuses on video action recognition tasks and presents CLAVER to improve CLIP by aligning video representations with verb understanding. The authors design the Kronecker mask attention for improved temporal modeling and interpretive prompts generated by large language models to enhance verb comprehension. The authors demonstrate that CLAVER shows competitive performance on benchmarks such as Kinetics-400, Kinetics-600, HMDB-51, and UCF-101.

**Strengths:**

1. The paper observes two challenges when applying CLIP to video action classification: How to perform effective temporal modeling and how to design suitable text descriptions for verb understanding. The authors then propose the Kronecker mask temporal attention (KMTA) and Kronecker mask causal temporal attention (KMCTA) to solve 1) and interpretive prompts to solve 2).
2. The paper conducts experiments on multiple video action classification datasets under supervised, few-shot, and zero-shot settings. The results show improved performance over baseline models and other state-of-the-art approaches.
3. The paper includes thorough ablation studies to isolate the contribution of each component, such as KMTA and interpretive prompts, providing evidence for their effectiveness.

**Weaknesses:**

1. My major concern about this work is its novelty. The proposed Kronecker Mask Attention seems to be a simple combination of pipeline temporal attention and joint attention by attending to all patches in other frames. I expect the authors to provide more insightful analysis and discussion about the proposed method and the differences from previous works.
2. The computational cost introduced by the Kronecker mask attention and the interpretive prompts could be further detailed, especially regarding training and inference time compared to other baselines.

**Questions:**

See weakness section

---

> ### Author Response · Authors · 2024-11-17
> **Author's Response (1/2)**
>
> We appreciate the reviewer’s thoughtful comments and constructive feedback. Below, we address each of the concerns raised.
>
> **1) The motivation and advantages of KMT, KMCT and the difference from previous method:**
>
> As illustrated in Fig. 2, our Kronecker Mask Temporal Attention (KMTA) and Kronecker Mask Causal Temporal Attention (KMCTA) are designed to overcome the limitations of existing temporal modeling approaches. Common methods such as pipeline temporal attention, alignment-guided temporal attention, and class-token-only temporal attention have limited temporal receptive fields, limiting their ability to capture dynamic information. Joint attention exhibits global receptive fields, on the other hand, faces the issue of spatiotemporal homogenization. Our KMTA leverages a Kronecker mask to simultaneously expand the temporal receptive field and mitigate spatiotemporal homogenization.
>
> **Spatiotemporal Homogenization**: We define this as the phenomenon where random token shuffling (i.e., destroying the spatiotemporal structure of video clips) has limited impact on word importance and similarity, which is counterintuitive. If there is a significant change in the visual branch's semantics, it will correspondingly cause changes in the word importance and similarity. This reflects whether spatiotemporal modeling captures interpretable video representations. Specifically, word importance refers to the correlation between each word in a sentence and the semantics of the video content, while similarity refers to the similarity between visual and textual representations.
>
> Fig. 6 illustrates the effects of token shuffling. Joint attention is minimally impacted, indicating spatiotemporal homogeneity. Conversely, KMT and KMCT are significantly affected, demonstrating their ability to capture interpretable spatiotemporal semantics. This highlights a key issue: learnable positional encoding has limited ability to mitigate spatiotemporal homogenization. Their optimization is driven by losses that aim to maximize accuracy, rather than explicitly learning spatiotemporal structures. In contrast, our Kronecker masks provide a natural inductive bias for spatiotemporal heterogeneity, leading to better interpretability. As shown in Appendix Fig. 8, previous state-of-the-art methods such as ILA and X-CLIP also encounter spatiotemporal homogenization in our experiments. Surprisingly, in some cases, token shuffling increases similarity, which is illogical.
>
> In Fig. 2 (Right), the spatial and temporal attention we mentioned both can be derived by combining a tailored Kronecker mask with joint attention. Therefore, we collectively refer to them as Kronecker Mask Attention. In addition, we also illustrate Fig. 10 as an example to show that Kronecker mask has certain generality, such as for spatiotemporal graph data modeling.
>
> **KMCTA’s Contribution**: KMCTA alleviates the low-rank bottleneck faced by KMTA. As the number of tokens increases (e.g., via higher frame lengths or reduced patch sizes), the expressive power of attention mechanisms will deteriorate, because the token count ($n$) exceed the dimension ($d$) of the token features ($n\gg d$). This leads to a low-rank self-attention matrix and feature homogenization. KMCTA ensures a full-rank self-attention matrix, as proven in the Appendix A. Tab. 8 demonstrates that JA and KMTA show limited improvements even degradations with increased frame lengths, while KMCTA consistently improves, underscoring its robustness in long frame scenarios.

---

> > ### Author Response · Authors · 2024-11-17
> > **Author's Response (2/2)**
> >
> > **2) About the computational cost, training and inference time:**
> >
> > We provide a detailed comparison of FLOPs, parameter size, and inference time for CLIP, ILA, XCLIP and our CLAVER in the table below. In addition, our model is trained on 4 NVIDIA A100 GPUs (CLAVER-B-32, CLAVER-B-16) or 8 NVIDIA A100 GPUs (CLAVER-L-14) for 3-5 days (on Kinetics-400 dataset). We test the inference time using a NVIDIA GeForce RTX 4090 GPU.
> >
> > The parameter size of CLAVER (B/32, B/16, L/14) are 1.3x that of CLIP (B/32, B/16, L/14), the values of FLOPs increase by 1.2x, and the increment of inference time are (1x, 2x, 3.6x) respectively. The parameter size of CLAVER (B/32, B/16, L/14) are very similar to X-CLIP (B/32, B/16, L/14), with FLOPs being 1.3x and inference time increments of (0.75x, 1.3x, 1.6x), respectively. In terms of performance, CLAVER is much better than X-CLIP. The parameter size of CLAVER (B/32, B/16, L/14) are 0.77x\~0.8x that of ILA (B/32, B/16, L/14), with FLOPs being 0.8x\~0.86x and inference time increments of (0.2x, 0.4x, 0.7x), respectively. CLAVER is better than ILA in both terms of performance and computational cost.
> >
> > | Model                    | FLOPs(G) | Params(M) | Inference time(ms) |
> > |----------------------|----------|-----------|--------|
> > | CLIP-B/32$_{8f}$   | 25.51    | 84.23     | 7.47  |
> > | CLIP-B/16$_{8f}$   | 92.10    | 82.46     | 8.81  |
> > | CLIP-L/14$_{8f}$   | 419.52  | 258.72   | 20.13 |
> > | X-CLIP-B/32$_{8f}$ | 25.64   | 99.72    | 9.54   |
> > | X-CLIP-B/16$_{8f}$ | 92.39   | 97.95    | 12.46 |
> > | X-CLIP-L/14$_{8f}$ | 420.38  | 302.87  | 46.39 |
> > | ILA-B/32$_{8f}$    | 40.24    | 133.08   | 33.65   |
> > | ILA-B/16$_{8f}$    | 150.59  | 131.30   | 36.89   |
> > | ILA-L/14$_{8f}$    | 647.70   | 395.65    | 104.03 |
> > | CLAVER-B/32$_{8f}$ | 33.06   | 103.11  | 7.18   |
> > | CLAVER-B/16$_{8f}$ | 121.85  | 101.35  | 16.42 |
> > | CLAVER-L/14$_{8f}$ | 557.50  | 325.87 | 72.81   |
> >
> > \* Note that the FLOPs here are calculated with a batch size of 1, 8 frames and a maximum sentence length of 77. The tools used to calculate these metrics are thop.profile and torch.utils.benchmark, where the inference time is the average of 100 runs.

---

> > > ### Comment · Reviewer_rNM4 · 2024-12-02
> > >
> > > Thanks to the authors for the responses about the contribution and computational efficiency. I keep my opinion that that the proposed Kronecker Mask Attention is a simple combinition of previous attention mechanisms to improve the receptive fields and naturally bring marginal improvement to the performance. I appreciate the authors' efforts on providing additional experiment on measuring the computational efficiency, so I'd like to raise my score to borderline accept.

---

> > > > ### Author Response · Authors · 2024-12-02
> > > > **Thank you for raising the score!**
> > > >
> > > > Thank you! We sincerely appreciate the time and effort you dedicated to reviewing our rebuttal.

---

### Author Response · Authors · 2024-11-20
**Summary: Reply to all reviewers**

We are deeply grateful to all the reviewers for their thoughtful consideration and constructive suggestions of our paper. We also appreciate their recognition of the novelty and the fluent writing. Based on the initial reviews, we have summarized their questions and provided corresponding responses, which mainly cover the following aspects:

+ **Detailed Comparison of Computation Cost and Efficiency** (proposed by reviewers rNM4 and 2oCA). According to the reviewer's request, we have listed a comprehensive comparison table.

+ **The Pipeline and Potential Effects of Interpretive Prompts** (noticed by reviewers RydQ and sCy3). We have provided detailed explanations and analysis under the comments of the relevant reviewers.

+ **Insightful Motivation and Analysis, Effectiveness of Our Spatiotemporal Modeling Approach** (highlighted by reviewer rNM4 and 2oCA). Please refer to the corresponding responses, as well as our emphasis on contributions below.

+ **More Representative Visualization of Conversion of Word Importance from Nouns to Verbs** (pointed by reviewer RydQ). We have provided Fig. 15 in the appendix of the rebuttal revision.

Again, we want to emphasize that this paper is dedicated to achieving three primary contributions:

+ Exploring **a General Spatiotemporal Modeling Approach** (**Kronecker Mask Attention**) and **Revealing the Intrinsic Correlations of Commonly Used Spatiotemporal Modeling Methods** (as mentioned in Sec. 3.1 and Fig. 2, most common spatiotemporal attention can be uniformly classified as Kronecker Mask Attention). We propose the Kronecker mask temporal attention (KMTA) for temporal modeling, aiming to capture the long-range and wide-range dependencies among frames with spatiotemporal heterogeneity. And further improve KMTA to Kronecker mask causal temporal attention (KMCTA) to alleviate the low-rank bottleneck of KMTA. Through multiple visualizations in Fig. 6-8, 12-14, we illustrate the interpretability of spatiotemporal modeling.

+ Proposing **a Potential Universal Tool for Understanding Abstract Verbs** (**Interpretive Prompts**), which encompasses action decomposition, synonym conversion, and collaboration with body parts. By doing so, it promotes the model's attention and understanding of action concepts, as demonstrated through visualizations in Fig. 5, 15. Additionally, it also contributes to zero-shot generalization and helps with the action recognition/understanding in open scenarios.

+ Integrate the above two techniques to efficiently **Shifting the Alignment in CLIP (VLMs) from Visual Objects and Nouns to Action Behaviors and Verbs**, tapping into the potential of VLMs in video understanding.

Furthermore, in accordance with the reviewers' suggestions, we have made several modifications to the article in the rebuttal revision.  Specifically, we update Fig. 4 and Fig. 6, as well as Tab. 8 and Tab. 14. Additionally, we provide more representative visualizations of the word importance in Fig. 15 and the rank of KMTA, KMCTA and joint attention with respect to various patch size and frame length in Fig. 16. The modified parts are marked in $\textcolor{blue}{\mathrm{blue}}$ font.

We are looking forward to having discussions with the reviewers in the next few days, focusing on the parts that might still be ambiguous in the rebuttal and other relevant issues. We sincerely welcome them to join in the discussion.

---

### Meta-Review · Area_Chair_qn8T · 2024-12-25

**Metareview:**

The submission addresses the problem of adapting contrastive language-*image* pretrained models (CLIP) to video domains. It introduces "Contrastive Language-Action Video Learner" (CLAVER), which adapts not only the visual branch (via "Kronecker Mask Attention" for temporal modeling) but also the text branch (via LLM-generated prompts that focus on verbs) of the CLIP model. After rebuttal, all four reviewers recommend acceptance of the submission, with one accept (8) and three borderline accepts (6). Most of the concerns were addressed by the authors' rebuttal, with minor remaining concerns on the novelty of the introduced "Kronecker Mask Attention" (rNM4). Reviewer RydQ had concerns regarding KMCTA, which the AC believes has been mostly addressed by the rebuttal. Overall, the AC agrees with the reviewers and believes the submission is ready to be published at ICLR 2025.

**Additional Comments On Reviewer Discussion:**

Please find above.

---

### Decision · Program_Chairs · 2025-01-22

Accept (Poster)